# Epigenetic regulation of serine biosynthesis by PHF8 during neurogenesis

Marta H Artes [1,10], Simona Iacobucci [1,10], María J Barallobre[1,2], Paula Carballeira [3], Marta Garcia-Cajide[4,5], Alejandro Pérez-Venteo[4,5], Natalia Padilla [6], Bárbara S Viegas [1], Aitana Díaz-Vásquez [6], A Silvina Nacht[7,8], Guillermo P Vicent [1], Maria L Arbonés[1,2], Xavier de la Cruz[6,9], Marta Nieto [3], Neus Agell [4,5], Caroline Mauvezin [4,5] & Marian A Martínez-Balbás [1✉]

## Abstract

**Progenitor proliferation during neurodevelopment requires tight coordination of epigenetic regulation and metabolism. However, the crosstalk between these processes remains poorly understood. To investigate this, we examine in neural stem cells the role of PHF8, a histone demethylase whose mutations are linked to Siderius-Hamel syndrome, a rare neurodevelopmental disorder. Through an integrated multi-omics approach - combining transcriptomics, epigenomics, and metabolomics - we identify PHF8 as a key driver of the serine biosynthesis pathway, safeguarding the intracellular serine pool essential for neural progenitor proliferation. PHF8 fine-tunes chromatin accessibility at promoters of metabolic genes, ensuring their activation during development. Loss of PHF8 disrupts amino acid metabolism, blocks autophagy, and hinders vesicle formation. Ultimately PHF8 depletion leads to replication defects, DNA damage, and proliferation arrest. In vivo, PHF8 deficiency in mouse embryos halts neurogenesis, progenitor expansion, and neuron generation in the developing brain. These findings identify PHF8 as a key molecular link between chromatin regulation, metabolic control, and neural development, offering new insights into the epigenetic basis of neurodevelopmental and metabolic disorders.**

**Keywords** PHF8; Gene Transcription; Serine Biosynthesis Histone Demethylation; Neural Stem Cells; Neurogenesis
**Subject Categories** Chromatin, Transcription & Genomics; Metabolism; Neuroscience

## Introduction

Over the past two decades, numerous chromatin modifiers have been recognized as key regulators of developmental processes. Notably, the activity of chromatin-modifying enzymes is profoundly influenced by metabolites, which serve as substrates, cofactors, or inhibitors. This dynamic interaction effectively transforms these enzymes into metabolic sensors and operate to translating cellular metabolic states into epigenetic reprogramming, positioning metabolism as a critical regulatory layer of the genome (Boon et al, 2020; Dai et al, 2020; Haws et al, 2020; Rabhi et al, 2017). Despite the emerging awareness of its importance, the relationship between metabolism and epigenetics remains largely unexplored, particularly within the context of neurodevelopment.

Developmental stem cell proliferation places considerable demands on both metabolism and epigenetic regulation (Intlekofer and Finley, 2019). This is especially true for the synthesis of nucleotides and biomass (Falkenberg et al, 2019; Hamalainen et al, 2019). A key example is one-carbon metabolism, which is essential for purine and pyrimidine biosynthesis, as well as serine production —processes that occur in parallel with extensive epigenomic remodeling driven by histone acetyltransferases (KATs), histone methyltransferases (KMTs), and histone demethylases (KDMs). Among the latter, PHF8, a JmjC-containing histone demethylase from the α-KG-dependent dioxygenase family, has emerged as an important factor in development. PHF8 specifically removes mono- and dimethyl groups from lysine 9 on histone H3 (H3K9me2) and monomethyl groups from lysine 20 on histone H4 (H4K20me1) (Fortschegger et al, 2010; Horton et al, 2010; Kleine-Kohlbrecher et al, 2010; Liu et al, 2010). Mutations in *PHF8* are linked to Siderius-Hamel syndrome, a rare X-linked intellectual disability (XLID) often accompanied by cleft lip and/or cleft palate (CL/P) (ORPHA:85287) (Abidi et al, 2007; Koivisto et al, 2007; Laumonnier et al, 2005; Siderius et al, 1999). Many of these

[1]Instituto de Biología Molecular de Barcelona (IBMB), Consejo Superior de Investigaciones Científicas (CSIC), Barcelona 08028, Spain. [2]Centro de Investigación Biomédica en Red de Enfermedades Raras (CIBERER), Instituto de Salud Carlos III, Barcelona, Spain. [3]Department of Molecular and Cellular Biology, Centro Nacional de Biotecnología, Consejo Superior de Investigaciones Científicas (CNB-CSIC), Madrid 28049, Spain. [4]Departament de Biomedicina, Facultat de Medicina i Ciències de la Salut, Universitat de Barcelona, Barcelona, Spain. [5]Institut d'Investigacions Biomèdiques August Pi i Sunyer (IDIBAPS), Barcelona, Spain. [6]Vall d'Hebron Institute of Research (VHIR), Passeig de la Vall d'Hebron, 119, E-08035 Barcelona, Spain. [7]Center for Genomic Regulation (CRG), Barcelona Institute for Science and Technology (BIST), Barcelona, Spain. [8]Universitat Pompeu Fabra (UPF), Barcelona, Spain. [9]Institut Català per la Recerca i Estudis Avançats (ICREA), Barcelona 08018, Spain. [10]These authors contributed equally: Marta H Artes, Simona Iacobucci. ✉E-mail: mmbbmc@ibmb.csic.es

mutations impair PHF8's catalytic activity (Loenarz et al, 2010; Qiu et al, 2010). Loss of PHF8 function disrupts neuronal differentiation, synapse formation, and cell survival, as demonstrated in various models ranging from mice to zebrafish (Qiu et al, 2010). Furthermore, PHF8 deficiency in mammals leads to impaired learning and memory, via alterations in the RSK-mTOR-S6K signaling pathway (Chen et al, 2018). *Phf8* knock-out (KO) mice also exhibit resistance to anxiety and depression-like behaviors, linked to the dysregulation of serotonin receptor expression (Htr1a and Htr2a) (Walsh et al, 2017). These findings strongly suggest that PHF8 plays a pivotal role in neurodevelopment.

In this study, we demonstrate that PHF8 is essential for neural stem cell proliferation in vitro and in vivo by orchestrating a metabolic program centered on amino acid biosynthesis, particularly serine production. Our findings reveal a novel mechanism by which PHF8 integrates cell renewal with metabolic demands, providing new insights into the interplay between epigenetics and metabolism during neurogenesis.

# Results

## PHF8 maintains neural stem cell proliferation

Analysis of publicly available single-cell RNA sequencing data from the developing mouse cerebral cortex (Di Bella et al, 2021; Data ref: Di Bella et al, 2021) revealed that *Phf8* is expressed in apical and intermediate progenitors at embryonic days (E)12 and E16 (Fig. 1A; Appendix Fig. S1A). Given that the embryonic cortex at these stages is largely composed of progenitors, we sought to elucidate the functional role of PHF8 during early corticogenesis. To this end, we employed a well-established self-renewal model using neural stem cells (NSCs) derived from E12.5 mouse embryonic cortices, which predominantly consist of progenitors, namely radial glial cells. These NSCs exhibit robust proliferative capacity and can differentiate into multiple neural lineages in vitro (Fig. 1B) (Currle et al, 2007; Estaras et al, 2012; Pappa et al, 2019; Pollard et al, 2006). To knock down PHF8 in NSCs, cells were transduced with lentiviral vectors carrying either control shRNA (shCTR) or PHF8-targeting shRNA (shPHF8). This efficiently reduced PHF8 mRNA and protein levels (Fig. 1B) without affecting the expression of the other two KDM7 subfamily members, Phf2 and Kiaa1718, which share both structural and functional similarities with PHF8 (Appendix Fig. S1B).

To analyze the role of PHF8 in NSCs, we first examined the consequences of its depletion on cell proliferation. shPHF8 NSCs exhibited a marked reduction in cell growth compared to shCTR cells (Fig. 1C), a phenotype also observed in other highly proliferative cell types, including cancer cell lines (Lim et al, 2013; Liu et al, 2010). Similar results were observed when a second independent shRNA that targets PHF8 (shPHF8-2) was used (Appendix Fig. S1C). Flow-cytometry analysis revealed a significant decrease in S-phase cells, both by propidium iodide (PI) staining and BrdU incorporation, and a parallel, although not significant, increase in % of G1 cells in shPHF8 cells. Moreover, while there was not a global significant change in G2 + M a reduction in the % of mitotic (MPM2 leveled) cells was observed in shPHF8 cells (Fig. 1D and Appendix Fig. S1D). These data suggest that, although cells were able to proliferate, they had defects in

progression into S phase and into M phase in the absence of PHF8. On the other hand, PHF8 overexpression did not affect cell proliferation (Appendix Fig. S1E).

## PHF8 regulates the serine biosynthesis pathway (SBP) gene transcription

To investigate the role of PHF8 in NSCs proliferation, we determined the PHF8-dependent transcriptional profile by performing RNA sequencing (RNA-seq) of control shCTR) and PHF8-depleted (shPHF8) NSCs samples in triplicate. The quality of the RNA-seq data was assessed by examining sample clustering using Pearson correlation analysis (Appendix Fig. S2A). Differential expression results were validated by quantitative polymerase chain reaction (qPCR) (Appendix Fig. S2B).

The analysis identified 5913 transcripts that exhibited significant changes in expression upon PHF8 knockdown [log2 fold change (FC) >0.5 and (FC) <−0.5 and *p*-value < 0.05] (Fig. 2A). Among these, 49.2% were downregulated and 50.7% were upregulated following PHF8 depletion (Fig. 2B). Notably, increasing the $\log_2$FC threshold to 1 did not substantially alter the relative proportions of up- and downregulated genes (Appendix Fig. S1C). These results are consistent with previous reports describing PHF8 as both a transcriptional activator and repressor (Asensio-Juan et al, 2017; Wang et al, 2014). Gene ontology (GO) enrichment analysis of the differentially expressed genes revealed the presence of categories associated with DNA replication, mitotic sister chromatid segregation, and chromatin remodeling at the centromere (Fig. 2C). These findings are in line with previous observations in cancer cell lines (Asensio-Juan et al, 2012; Liu et al, 2010). Interestingly, the most enriched category was related to metabolism (Fig. 2C). Further classification of the differentially expressed metabolic genes indicated that serine metabolism was the most enriched among the metabolic processes (Fig. 2D).

Next, we focused on genes within the serine biosynthesis pathway (SBP) that were significantly regulated in the RNA-seq dataset. The majority of SBP-associated transcripts were downregulated in PHF8-depleted NSCs (Fig. 2E). qPCR validation confirmed the reduced expression of the three key SBP enzymes—Phgdh, Psat1, and Psph—in shPHF8 NSCs compared to control cells (shCTR) (Fig. 2F). Notably, doxycycline-induced overexpression of PHF8 from the *pInducer-PHF8* construct restored the transcription of these SBP genes to control levels (Fig. 2F). Collectively, these results demonstrate that PHF8 acts as a crucial transcriptional regulator of the serine biosynthesis pathway in NSCs.

The SBP is particularly activated under serine-deprived conditions. To determine whether PHF8 is required for this adaptive response, we examined the transcriptional induction of *Phgdh*, *Psat1*, and *Psph* in shCTR and shPHF8 NSCs cultured in serine-free medium [medium lacking non-essential amino acids (NEAAs)]. As shown in Fig. 2G, the expression of SBP genes was upregulated in shCTR cells but not in PHF8-depleted cells, indicating that PHF8 is required for their proper induction under serine starvation. These findings suggest that PHF8 is essential for maintaining SBP activity. Consistent with this, PHF8-depleted NSCs exhibited increased sensitivity to NEAA deprivation. NEAA depletion impaired proliferation in shPHF8 cells, but not in shCTR NSCs, further supporting a role for PHF8 in sustaining amino acid homeostasis required for NSC proliferation (Fig. 2H). In agreement, treatment with a non-NAD + -competing allosteric PHGDH inhibitor caused

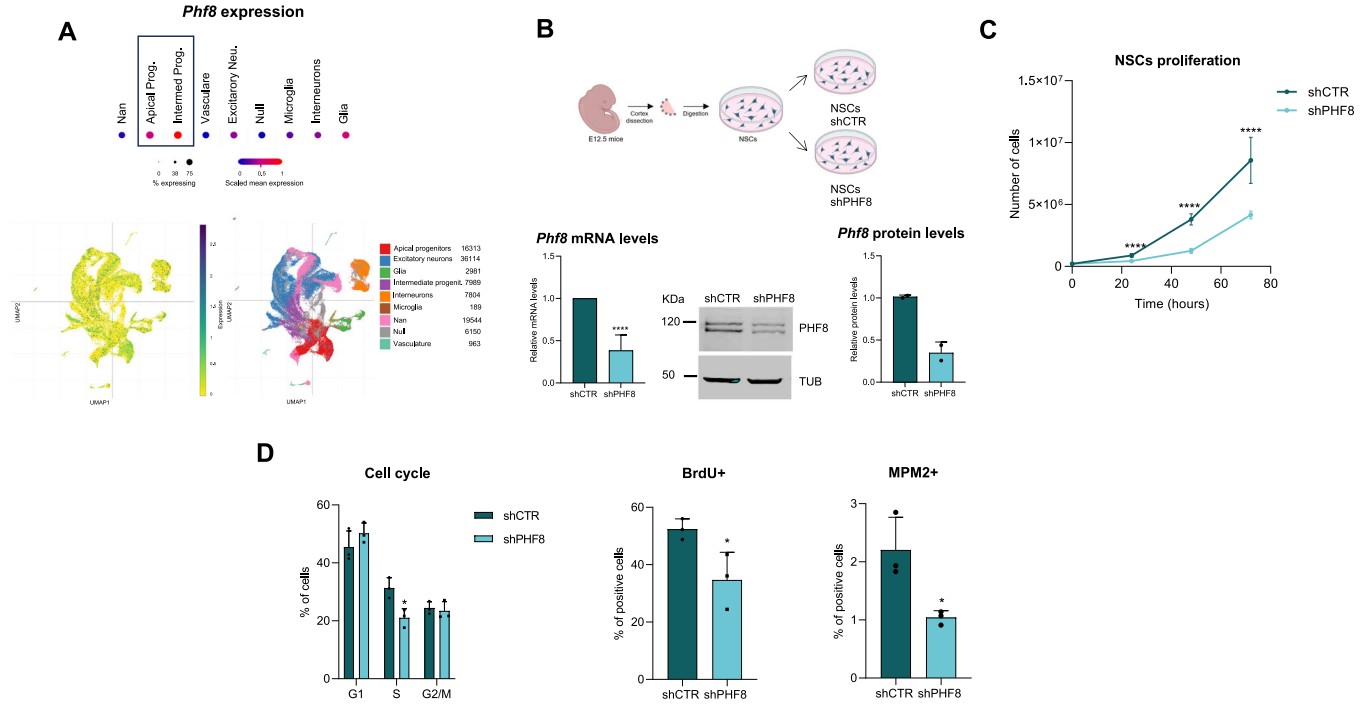

**Figure 1. PHF8 maintains neural stem cell proliferation.**

(A) UMAP plot showing *Phf8* expression in the developing mouse cerebral cortex. Data were derived from a publicly available single-cell RNA-seq dataset (Di Bella et al, 2021; Data ref: Di Bella et al, 2021). (B) Schematic representation of the experimental model used in this study. Neural stem cells (NSCs) were dissected from cerebral cortices of mouse fetal brains (E12.5) and cultured ex vivo (see methods). NSCs were infected with lentivirus expressing shRNA control (shCTR) or PHF8-specific shRNA (shPHF8). After 24 h, shRNA-expressing cells were selected with puromycin. 48 h post-infection, total RNA and protein extracts were collected to assess PHF8 mRNA ($n = 6$ biologically independent experiments performed in triplicate) and protein levels via qPCR and immunoblotting ($n = 2$), respectively. mRNA expression values were normalized to the housekeeping gene *Gapdh*, and protein levels were normalized to tubulin. Error bars represent the mean with standard deviation (SD). ****$p < 0.0001$, two-tailed Student's *t*-test. (C) Growth curve showing proliferation of NSCs infected with shCTR or shPHF8 lentivirus over 72 h. Data represent the mean of eight biologically independent experiments performed in triplicate. Error bars represent the mean with SD. ****$p < 0.0001$, two-tailed Student's *t*-test. (D) Graphs showing flow-cytometry quantification of shCTR and shPHF8 NSCs. The left graph corresponds to PI staining quantification (*$p = 0.0194$), the middle graph to BrdU incorporation quantification relative to total cells (*$p = 0.0410$), and the right graph displays MPM2 levels relative to total cells (*$p = 0.0247$, two-tailed *t*-test). Data represent the mean of three biologically independent experiments. Error bars represent the mean with SD. Original flow-cytometry data are presented in Appendix Fig. S1D. Source data are available online for this figure.

---

a delay in NSCs proliferation (Appendix Fig. S2D). Finally, closer inspection of the RNA-seq dataset revealed that, in addition to SBP genes, several amino acid transporter transcripts were also dysregulated in PHF8-depleted NSCs compared with controls (Appendix Fig. S2E).

## PHF8 maintains low levels of H4K20me1 at SBP gene promoters

To test whether PHF8 directly regulates the metabolic enzymes involved in serine biosynthesis, we analyzed previously published PHF8 ChIP-seq data (Appendix Table S1). We compared PHF8 binding sites with histone modifications associated with active (H3K4me3) and repressive (H4K20me1, H3K9me2) chromatin states. PHF8 was found to bind the transcription start sites (TSS) of *Phgdh*, *Psat1*, and *Psph*, where it colocalized with high levels of H3K4me3—a mark recognized by PHF8—and low levels of H4K20me1 and H3K9me2, which are repressive marks targeted by PHF8 (Fig. 3A). Although some of the ChIP-seq data used (Appendix Table S1) were derived from non-NSC cell types, raising

the possibility of cell type–specific effects, these results support the notion that PHF8 may directly bind to SBP gene promoters and demethylate them to facilitate transcription.

To further understand how PHF8 regulates SBP gene expression, we assessed the effects of PHF8 depletion (Fig. EV1A) on histone methylation in NSCs. Immunofluorescence analysis of known PHF8 histone targets—H4K20me1 and H3K9me2—revealed a marked global increase in H4K20me1 levels following PHF8 knockdown (Fig. EV1B), whereas H3K9me2 levels remained largely unchanged (Fig. EV1C). Motivated by the observed global increase in H4K20me1, we next conducted H4K20me1 ChIP-seq assays in shCTR and shPHF8 NSCs (Appendix Table S2; Appendix Fig. S3A). The ChIP-seq results were further validated by qPCR (Appendix Fig. S3B). We first examined the global changes in H4K20 chromatin monomethylation by segmenting the genome into 30Kb bins. This analysis revealed that PHF8 knock-down led to a higher number of genomic regions (bins) with increased H4K20me1 signal intensity, as compared to regions with decreased signal strength, relative to control cells (Fig. 3B). GO analysis of genes lying nearest to the H4K20me1 peak region and exhibiting increased H4K20me1 enrichment indicated their

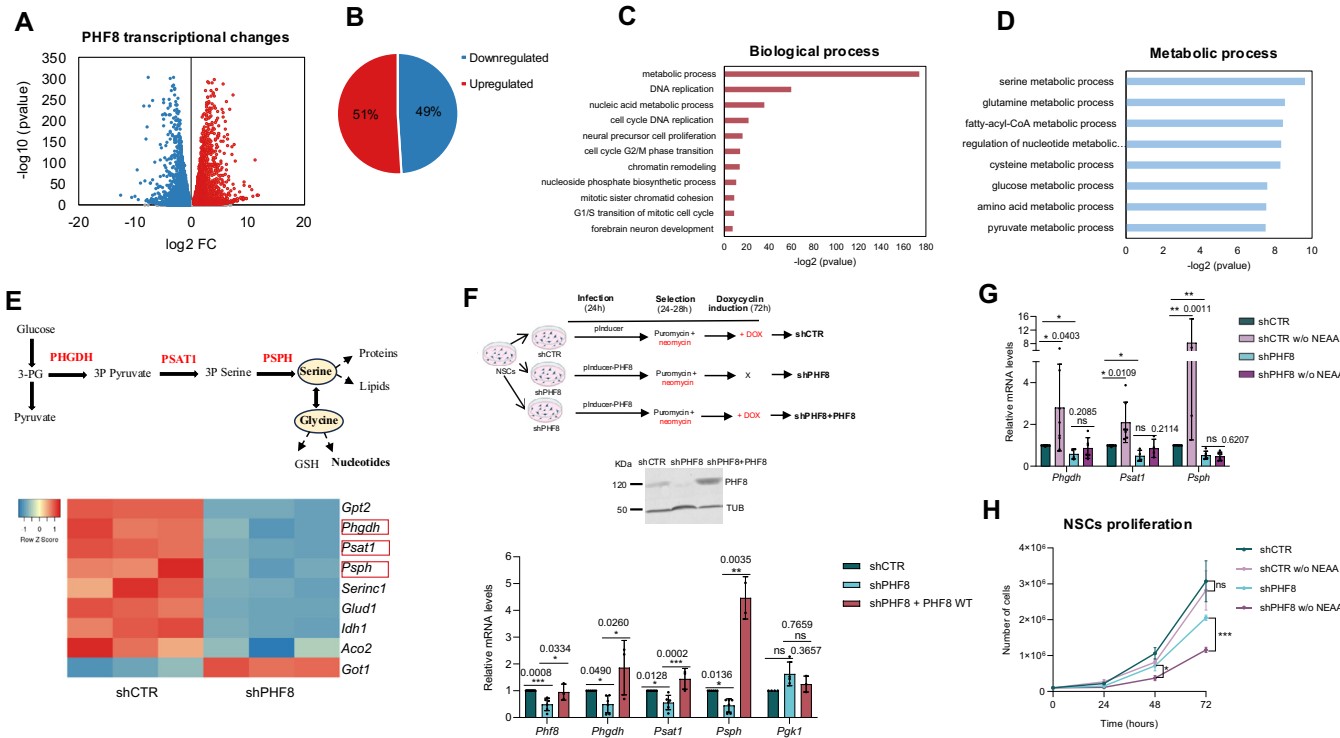

**Figure 2. PHF8 regulates transcription of Serine Biosynthesis Pathway (SBP) genes.**

(A) Volcano plot showing PHF8 transcriptional targets identified by RNA-seq in shCTR and shPHF8 NSCs. The red dots represent the genes with *p* value <0.05 and log$_2$ fold change >0.5; blue dots indicate genes with *p* value <0.05 and log$_2$ fold change <−0.5. Data used for this analysis correspond to the values obtained from three biologically independent RNA-seq replicates. Differential expression analysis was performed by BGI using the Dr. Tom RNA-seq analysis pipeline. Statistical significance was assessed using DESeq2; genes with an adjusted *p* value (*q* value) ≤0.05 were considered significant. DESeq2 uses a Wald test. (B) Graph depicting the percentage of upregulated and downregulated genes in shPHF8 compared to shCTR NSCs (*p* value <0.05). (C) Gene ontology (GO) analysis of Biological Process enriched among PHF8-regulated genes (*p* value <0.05 and log$_2$ fold change >0.5 and log$_2$ fold change <−0.5). The whole *Mus musculus* genome 10 (mm10) was used as background. (D) GO analysis highlighting metabolic processes among genes differentially regulated in the RNA-seq of shCTR and shPHF8 NSCs (*p* value <0.05 and log$_2$ fold change >0.5 and log$_2$ fold change <−0.5). (E) Top panel: Schematic diagram of the serine biosynthesis pathway with key enzymatic steps. Bottom panel: Heatmap showing expression of selected serine metabolism-related genes in shCTR and shPHF8 NSCs from RNA-seq data. All genes shown meet the criteria: *p* value <0.05 and log$_2$ fold change >0.5 or < −0.5. (F) PHF8 was induced in shPHF8 NSCs (immunoblot shown at the top of the panel), and expression levels of the indicated genes were measured by qPCR. Expression values were normalized to the housekeeping gene *Gapdh* and presented relative to shCTR NSCs. *Pgk1* served as a negative control. *n* = 3–6 biologically independent experiments were performed. Error bars represent the mean with SD. Two-tailed *t*-test was applied. (G) shCTR and shPHF8 NSCs cells were cultured in media with or without (w/o) non-essential amino acids (NEAA) for 6 h. Expression of *Phf8*, *Phgdh*, and *Psat1* was measured by qPCR, normalized to *Gapdh*. Data were derived from 4 to 7 biologically independent experiments. Normality was confirmed using the Shapiro–Wilk test, and statistical significance was assessed with a two-tailed *t*-test. Error bars represent the mean with SD. (H) Growth curve showing proliferation of shCTR and shPHF8 NSCs in the presence or absence of NEAA over 0 to 72 h. Data used for this analysis correspond to the values obtained from three biologically independent experiments. Error bars represent mean ± SEM. ns, *p* = 0.5768; **p* = 0.0192; ****p* = 0.0001, two-tailed *t*-test. Source data are available online for this figure.

involvement in cellular metabolic processes, DNA transcription, and other fundamental processes related to cell proliferation and gene expression (Appendix Fig. S3C). These findings align with an overall increase in H4K20me1 bulk methylation upon PHF8 depletion (Iacobucci et al, 2021; Fig. EV1B). Finally, comparison between genes that gained H4K20me1 and RNA-seq revealed that 862 genes that were downregulated upon PHF8 depletion gained H4K20me1 at their promoters (Fig. 3C). Among these, some SBP genes were identified (Fig. 3C; Appendix Fig. S3D). These data suggest that PHF8 facilitates SBP gene transcription by demethylating their promoters. Interestingly, we observed a decrease in H4K20me1 levels at the SBP gene bodies upon PHF8 depletion, as expected due to downregulation of these genes (Appendix Fig. S3D), given that H4K20me1 enrichment along the gene body is associated with transcriptional activation and elongation (Shoaib et al, 2021).

To understand the lack of changes in H3K9me2 levels, and given that H3K9me2 serves as a substrate for Suv39h1/2, which catalyze its conversion to H3K9me3, we reasoned that the H3K9me2 generated after PHF8 loss might be further methylated, leading to higher H3K9me3 levels. To test this idea, we first analyzed global H3K9me3 levels (Fig. EV1D) and observed an overall increase. To further investigate H3K9me3 changes due to PHF8 depletion, we performed H3K9me3 ChIP-seq in shCTR and shPHF8 NSCs (Appendix Table S2; Appendix Fig. S4A). We initially examined global changes by segmenting the genome into 30 kb bins. This analysis revealed that PHF8 knockdown resulted in a greater number of genomic regions with increased H3K9me3 signal compared to regions with decreased signal relative to control cells (Appendix Fig. S4B). Further analysis indicated that genomic segments with decreased H3K9me3 were significantly enriched at

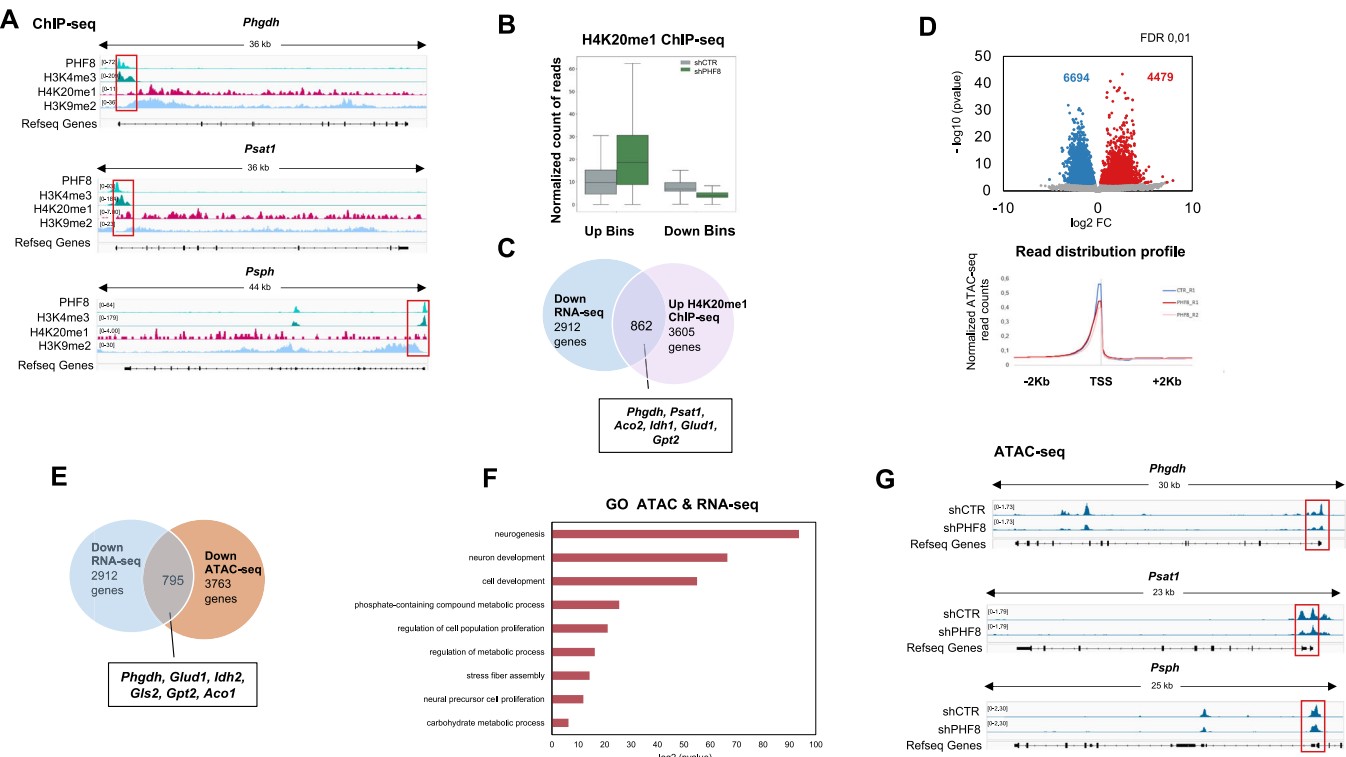

**Figure 3. PHF8 maintains competent chromatin for transcription at SBP gene promoters.**

(A) Integrated genome viewer (IGV) screenshots showing PHF8 binding peaks (input-subtracted) and ChIP-seq profiles for H3K4me3, H4K20me1 and H3K9me2 ChIP-seq at *Phgdh*, *Psat1*, and *Psph* loci in human embryonic stem cells (hESCs). Homo sapiens genome hg19. (B) H4K20me1 ChIP-seq was performed in duplicate in shPHF8 and shCTR NSCs. The mouse genome was divided into 180,000 bins of 30 Kb each to assess H4K20me1 enrichment. Box plots display bins showing gain and loss of H4K20me1 in shPHF8 cells. Box plot elements: centerline = median; box limits = 25th and 75th percentiles; whiskers = minimum and maximum. (C) Venn diagrams were generated to visualize the overlap between genes that gain H4K20me1 upon PHF8 depletion and are downregulated in RNA-seq (log2 fold change <0.05 and <−0.5; p value <0.05). No statistical test was applied to the Venn diagram itself, as it is a descriptive visualization. (D) ATAC-seq was performed in shPHF8 and shCTR NSCs. Peaks were identified as described in Methods, and fold change values were computed. The volcano plot displays peaks with increased (red; n = 4.479) or decreased (blue; n = 6.694) ATAC signal in shPHF8 cells. Differential chromatin accessibility analysis was performed using DESeq2. Statistical significance was assessed using the Wald test. Peaks with p value <0.01 and |log2 fold change|>0.5 were considered differentially accessible. Bottom panel: ATAC-seq read density plotted ±2000 bp from transcription start sites (TSS) in shCTR and shPHF8 NSCs. (E) Venn diagram showing the overlap between genes that lose chromatin accessibility upon PHF8 depletion and genes downregulated in PHF8 RNA-seq analysis. (F) Gene Ontology (GO) analysis of Biological Processes enriched among genes that are both regulated by PHF8 (log2 fold change >0.5 or <−0.5; p < 0.05) and show changes in chromatin accessibility upon PHF8 depletion (FDR 0.01). Enrichment was assessed using Fisher's exact test, and p values were corrected using the g:SCS method implemented in g:Profiler. (G) IGV screenshots showing ATAC signal profiles at *Phgdh*, *Psat1*, and *Psph* loci in shCTR and shPHF8 NSCs.

promoters, particularly at transcription start sites (TSS), whereas regions that gained H3K9me3 upon PHF8 depletion were predominantly intergenic (Appendix Fig. S4C). GO analysis of genes associated with increased H3K9me3 did not reveal enrichment for metabolic pathways, but rather for neurogenesis and neural function (Appendix Fig. S4D). A close examination of SBP gene promoters showed no changes in H3K9me3 levels following PHF8 depletion (Appendix Fig. S4E).

Collectively, these findings suggest that H4K20me1 plays a role in the PHF8-dependent regulation of SBP genes.

## PHF8 maintains competent chromatin for transcription at SBP gene promoters

We next conducted ATAC-seq assays in both control and two replicates of depleted PHF8 NSCs. This assay assesses chromatin accessibility and provides insights into the changes associated with PHF8 function. We found that the number of regions exhibiting decreased accessibility upon PHF8 depletion was higher than those showing increased accessibility (Fig. 3D). These data were consistent with the enhanced H4K20me1 and H3K9me3 levels observed upon PHF8 depletion in NSCs (Fig. EV1B,D). Further analysis demonstrated that regions that lost accessibility were significantly enriched in promoters, particularly at TSS (Fig. 3D, bottom panel; Appendix Fig. S4F). On the other hand, the regions that gained chromatin accessibility were primarily located at intergenic regions (Appendix Fig. S4F). Interestingly, GO analysis of the 795 genes exhibiting both reduced chromatin accessibility and downregulation in the RNA-seq dataset (Fig. 3E) identified a significant enrichment for genes related to neurodevelopment, cell proliferation, and metabolism (Fig. 3F), including SBP genes (Fig. 3G).

These data strongly suggest that PHF8 is responsible for maintaining chromatin accessibility at these specific sites, probably

by keeping low levels of H4K20me1. These findings reinforce the notion that PHF8 maintains chromatin competency for transcription at gene promoters that are crucial for the regulation of metabolic gene transcription.

## PHF8 cooperates with transcription factors to regulate SBP gene transcription

Next, we sought to investigate the mechanisms that target PHF8 to the SBP gene promoters. We analyzed the promoter sequences of metabolic genes whose expression is affected by the depletion of PHF8 (identified in the RNA-seq) using the Pscan tool (Zambelli et al, 2009), which revealed binding sites for NRF1, ATF4, E2F4, and NFIA (Fig. EV2A).

Analysis of single-cell RNA sequencing data from the developing mouse cerebral cortex (Di Bella et al, 2021; Data ref: Di Bella et al, 2021) indicated that, among these factors, NFIA, followed by ATF4, is the most highly expressed transcription factors and is present in the largest number of cells at developmental stages in which PHF8 is expressed (Fig. EV2B). Furthermore, comparison of PHF8 binding sites with those of NFIA and ATF4 at SBP gene promoters, using previously published ChIP-seq datasets, revealed colocalization of PHF8 and NFIA at the *Phgdh* promoter, and of PHF8 and ATF4 at the *Psat1* promoter (Fig. EV2C). These findings suggest that PHF8 may cooperate with these transcription factors to regulate gene expression, particularly in the control of serine biosynthesis.

Based on these observations, and given that we have previously shown that PHF8 regulates *Nfia* expression in different contexts (Iacobucci et al, 2021), we next investigated whether NFIA contributes to PHF8-mediated transcriptional regulation of *SBP* genes. NFIA is a member of the NFI family that plays critical roles in embryogenesis, including neocortical, hippocampal, retinal, and cerebellar development (Clark et al, 2019; Harris et al, 2016; Piper et al, 2014). We first evaluated the impact of NFIA depletion on the expression levels of the major enzymes of the SBP. To do so, NSCs were transduced with lentivirus containing the specific *Nfia* shRNA that efficiently decreased NFIA levels (shNFIA) (Fig. EV2D) without affecting the *Phf8* (Fig. EV2D). The transcript levels of *Phgdh*, *Psat1*, and *Psph* were subsequently analyzed, revealing a significant reduction in SBP gene transcription following NFIA depletion (Fig. EV2D). At the genome-wide level, analysis of previously published NFIA RNA-seq data (Appendix Table S1) revealed that 614 genes downregulated in PHF8 KD RNA-seq were also downregulated upon NFIA knockdown (Fig. EV2E). Notably, the number of genes co-regulated by PHF8 and NFIA was significantly higher than that observed when compared to 2048 randomly selected genes (corresponding to the number of genes downregulated in NFIA KD) (Fig. EV2E, right panel). GO analysis of these overlapping genes highlighted functional categories related to cell cycle progression, metabolism, and neural development (Fig. EV2F). Although some RNA-seq datasets were derived from non-NSC cell types—raising the possibility of cell type–specific effects—these findings support the notion that PHF8 and NFIA may cooperate to regulate SBP gene transcription. Finally, we examined the effect of NFIA depletion on NSC proliferation. shNFIA cells exhibited a marked decrease in growth at 72 h (Fig. EV2G), comparable to or even greater than that observed upon PHF8

depletion, and this effect was independent of PHF8 levels (Appendix Fig. S5A).

To evaluate the contribution of ATF4, NSCs were transduced with an ATF4-specific shRNA (shATF4), which efficiently reduced ATF4 expression (Appendix Fig. S5A). ATF4 knockdown caused only minor alterations in SBP gene expression—affecting primarily *Psat1* (Appendix Fig. S5B)—and had modest effects on NSC proliferation (Appendix Fig. S5C). Genome-wide analysis using previously published ATF4 RNA.seq (Appendix Table S1) identified 310 genes downregulated upon PHF8 depletion were also affected by ATF4 knockdown (Appendix Fig. S5D). These were significantly enriched for functional categories related to cell cycle progression and metabolism (Appendix Fig. S5E).

Collectively, these findings indicate that PHF8 cooperates predominantly with NFIA, and to a lesser extent with ATF4, during early corticogenesis to regulate the transcription of key enzymes involved in serine biosynthesis.

## PHF8 depletion impairs serine biosynthesis

As we observed that the main enzymes related to SBP were downregulated in shPHF8 NSCs (Fig. 2), we decided to evaluate the concentration of serine in PHF8-depleted cells. In order to do that, we performed liquid chromatography−mass spectrometry (LC/MS) experiments using three biologically independent replicates. The results in Fig. 4A showed a reduction in serine (Ser) concentration in shPHF8 compared to shCTR NSCs, as well as other amino acids (Appendix Fig. S6A). Knowing that shPHF8 NSCs suffer from serine depletion, which is essential for proliferation, we hypothesized that serine supplementation could rescue the growth defects of shPHF8 cells. Thus, we supplemented the culture media of both shCTR and shPHF8 cells with serine. While serine had no effect on shCTR cells, it enhanced the growth of shPHF8 cells (Fig. 4B)

Although serine is not directly incorporated into nitrogenous bases, it contributes to nucleotide synthesis in two key ways. First, it serves as a precursor to glycine, which is directly incorporated into purine formation. Additionally, when converted to glycine, serine donates one-carbon units which are essential for both purine ring and the conversion of dUMP to dTMP in pyrimidine synthesis (Fig. 4C) Thus, serine supports cellular nucleotide pools in an indirect yet indispensable manner, particularly in rapidly proliferating cells. A close analysis of the results showed that indeed glycine levels (Fig. 4D), as well as the nucleotides (Fig. 4D), were reduced in shPHF8 NSCs. The lack of nucleotides could lead to alterations in replication fork progression during the S phase of the cell cycle. To test this hypothesis, we performed DNA fiber assay to measure replication fork dynamics and stalling in shCTR and shPHF8 NSCs, which showed an increase in the percentage of stalled forks and a decrease in fiber length in PHF8-depleted cells (Fig. 4E; Appendix Fig. S6B). Alterations in replication fork progression are frequently associated with DNA damage, in particular with double-strand breaks (DSB), which we measured by the γH2AX content. The histone variant H2AX is phosphorylated at the Ser-139 residue, forming γH2AX (P-H2AX) as an early cellular response to the induction of DNA DSBs. Analysis of γH2Ax content as a measure of DNA damage in control and PHF8-depleted NSCs showed a significant accumulation of γH2AX upon PHF8 depletion that was rescued through serine supplementation in the culture medium (Fig. 4F).

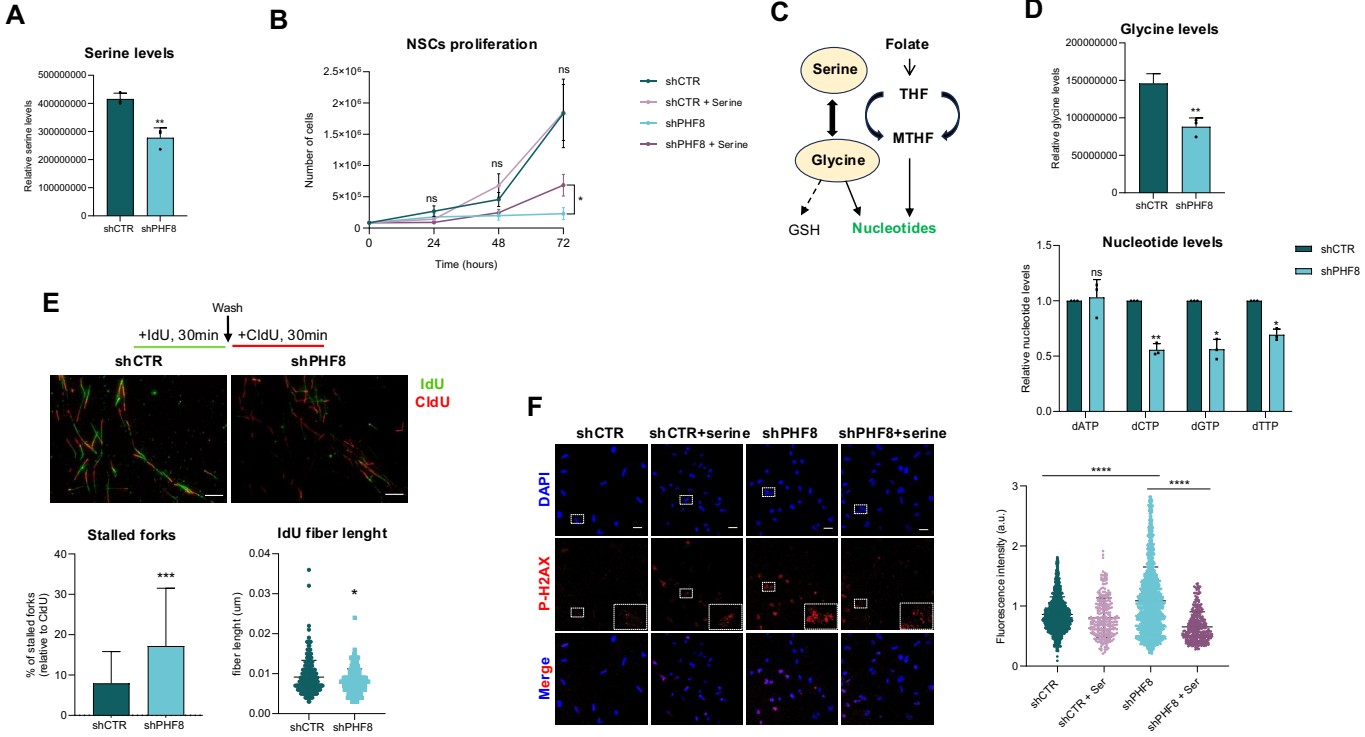

**Figure 4. PHF8 depletion impairs serine biosynthesis.**

(A) Metabolites from shCTR and shPHF8 NSCs were analysed by liquid chromatography−mass spectrometry (LC/MS). The quantification of serine levels from three biologically independent replicates of shCTR and shPHF8 samples is shown. Error bars represent mean ± SEM. **$p = 0.0043$, two-tailed $t$-test. (B) Growth curve showing the proliferation rate of shCTR and shPHF8 NSCs cultured in media supplemented or not with serine (5 mM) over a period of 0 to 72 h. Data represent the mean of six biologically independent experiments performed in triplicated. Error bars represent mean with SEM. *$p = 0.0392$; ns not significant, two-tailed $t$-test. (C, D) Quantification of glycine (C), and deoxynucleotides (dATP, dCTP, dGTP, and dTTP) (D) levels in shCTR and shPHF8 NSCs determined by LC/MS from three biological independent replicates. Error bars represent mean ± SEM. Glycine (**$p = 0.0046$); dATP (ns $p = 0.8427$), dCTP (**$p = 0.0012$), dGTP (*$p = 0.0345$), dTTP (*$p = 0.0298$), two-tailed $t$-test. (E) DNA fiber assay on shCTR and shPHF8 NSCs. Cells were sequentially labeled with IdU (30 min, red) followed by CldU (30 min, green). Graphs show % of stalled forks ($n = 50$; ***$p = 0.0004$, two-tailed Mann–Whitney test) and the fiber length quantification ($n = 200$; *$p = 0.0240$, two-tailed Mann–Whitney test). Data shown are representative of three biologically independent experiments. Error bars represent mean ± SEM. Scale bar: 20 μm. (F) Immunostaining of shCTR and shPHF8 NSCs cultured in normal medium or medium supplemented with serine (5 mM) using anti–P-H2Ax antibodies and DAPI. Data were representative of three biologically independent experiments. Scale bar: 20 μm. Violin plots show quantification of fluorescence intensity per cell, ($n = 289$–1699). Error bars represent mean ± SEM. ****$p < 0.0001$, two-tailed Mann–Whitney test. Source data are available online for this figure.

Altogether, these data demonstrate that PHF8 deficiency disrupts metabolite synthesis, leading to impaired replication and ultimately resulting in DNA damage.

## PHF8 depletion disrupts autophagy

Given that PHF8 depletion results in defects in amino acid synthesis, ultimately leading to cell cycle abnormalities, we investigated whether the loss of PHF8 triggers autophagy—a catabolic process activated to mitigate starvation-induced stress and maintain cellular homeostasis (He, 2022). To evaluate autophagic activity, we assessed autophagic flux by immunoblotting for LC3 lipidation and analysing the formation of autophagic vesicles, indicated by LC3-positive puncta, a common marker of autophagosome formation. LC3, the mammalian homolog of Atg8, is processed by the Atg4 protease and subsequently lipidated, enabling its incorporation into membranes of the autophagic vesicles and redistribution from a diffuse form (LC3-I) to a punctate pattern (LC3-II) (Klionsky et al, 2008; Mizushima et al,

2010). Unexpectedly, PHF8-depleted cells exhibited a significant accumulation of LC3-I protein without a corresponding increase in LC3-II or SQSTM1 (also known as p62) levels under basal conditions (Fig. 5A; Appendix Fig. S7A). Treatment with concanamycin A, which inhibits lysosomal v-ATPase, thereby preventing lysosomal acidification and degradation of autophagic cargo, led to the expected accumulation of LC3-II in control cells (Fig. 5A). In contrast, PHF8-depleted cells failed to exhibit this increase, suggesting an impairment in autophagic flux (Fig. 5A). Immunofluorescence analysis further supported these findings, revealing elevated LC3-I intensity in PHF8-depleted cells compared to control cells (Fig. 5B). As expected, concanamycin A markedly increased the number and size of LC3-II–positive autophagic vesicles in control cells. However, in PHF8-depleted cells, the number of LC3-II puncta was reduced, with no change in puncta size (Fig. 5B). These results suggest a defect in autophagic vesicle formation in cells depleted for PHF8. Furthermore, although concanamycin A increased vesicle number and size in PHF8-depleted cells, this effect was significantly attenuated compared to

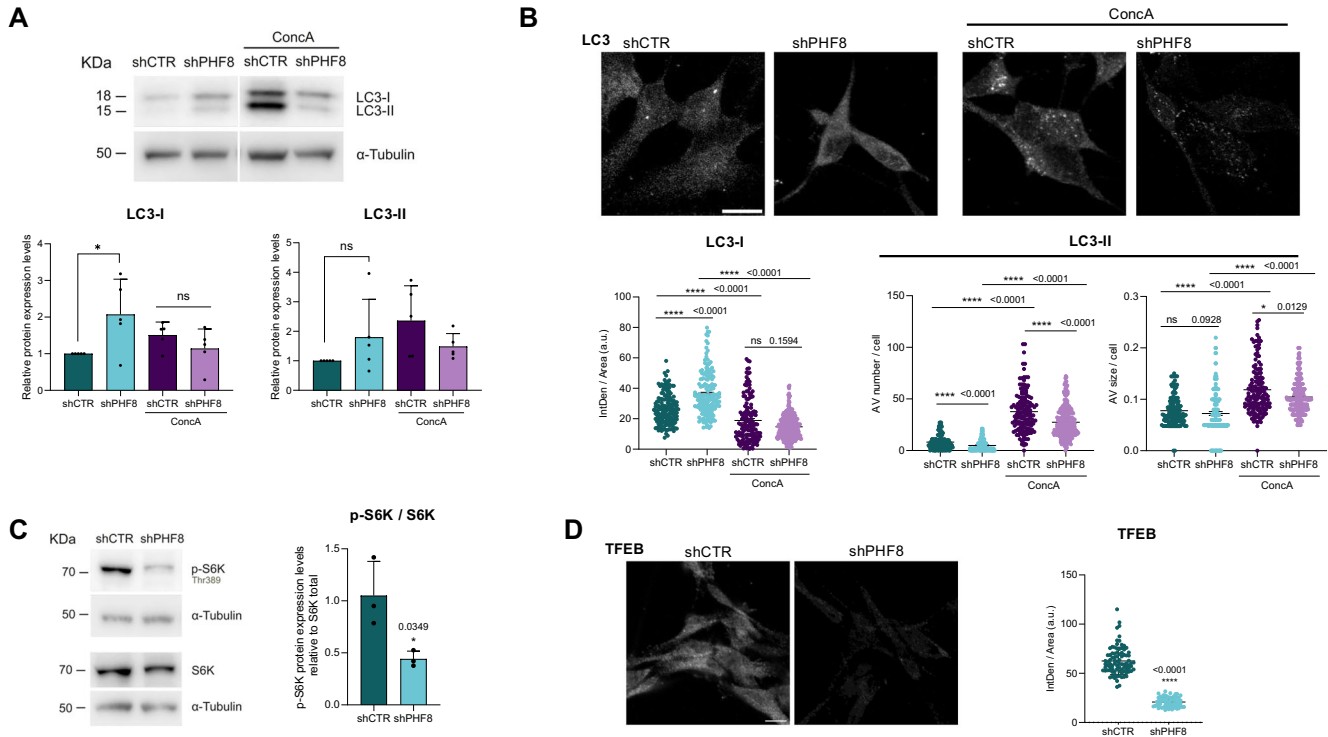

**Figure 5. PHF8 depletion disrupts autophagy.**

(A) shCTR and shPHF8 NSCs were treated or not with concanamycin A for 5 h. Total protein extracts were prepared and the LC3-I, LC3-II, and TUBULIN—as loading control levels were assessed by immunoblot. The samples derive from the same experiment and processed in the same blot. Graphs represent protein expression levels relative to tubulin. Data shown are representative of five biologically independent experiments. Error bars represent mean ± SEM. LC3-I. *$p = 0.0376$; ns not significant, two-tailed *t*-test. (B) Immunostaining assays of shCTR and shPHF8 NSCs treated or not with concanamycin A for 4 h. Cells were fixed and stained using anti LC3 antibody. Confocal images were acquired and total fluorescence intensity per cell was quantified as a measure of LC3-I levels. Additionally, the number and size of fluorescent foci were analysed to assess LC3-II ($n = 150$ cells). The data were representative of three biologically independent experiments. Scale bar indicates 10 µm. Error bars represent mean ± SEM. Two-tailed Mann–Whitney test was applied. (C) Immunoblot showing S6K and phosphorylated S6K (p-S6K) levels in shCTR and shPHF8 NSCs. p-S6K levels were quantified and normalized to total S6K. The data were representative of three biologically independent experiments. Error bars represent the mean with SD. *$p = 0.0349$, two-tailed *t*-test. (D) Immunostaining of endogenous TFEB in shCTR and shPHF8 NSCs using anti-TFEB antibody. The data shown are representative of three biologically independent experiments. Scale bar, 10 µm. Error bars represent the mean with SD. Violin plots represent quantification of the fluorescence intensity per cell ($n = 64$). ****$p < 0.0001$, two-tailed *t*-test. Source data are available online for this figure.

controls, reinforcing the conclusion that PHF8 loss impairs autophagic flux (Fig. 5B).

To further understand this impairment, we analysed the status of mTORC1, a master regulator of autophagy. Immunoblotting for S6K phosphorylation at Thr389, and ULK phosphorylation at Ser 757, downstream targets of mTORC1, revealed a significant reduction of mTORC1 signaling in PHF8-depleted cells (Fig. 5C; Appendix Fig. S7B). This finding is unexpected, as reduced mTOR activity typically promotes autophagy. Thus, our data suggest that both anabolic and catabolic pathways are disrupted in the absence of PHF8. RNA-seq analysis provided further insight, revealing altered expression of numerous autophagy-related genes, including regulators of the mTORC1 complex such as *Deptor* and *Pten* (Appendix Fig. S7C). Moreover, the protein levels of the transcription factor TFEB, which is critical for autophagy and lysosomal biogenesis (Perera et al, 2019), were reduced in PHF8-deficient cells compared to controls (Fig. 5D).

Taken together, these results underscore a critical role for PHF8 in maintaining metabolic homeostasis. PHF8 depletion not only

disrupts amino acid metabolism and mTORC1 signaling but also impairs autophagic flux, autophagic vesicle formation, and possibly TFEB-mediated transcriptional responses, further exacerbating metabolic stress.

## PHF8 depletion reduces mouse neurogenesis in vivo and impairs the differentiation of the neuronal outputs

Our data indicate that PHF8 activates genes critical for the metabolic support of neural progenitor proliferation. To assess the impact of PHF8 depletion in cortical neurogenesis in vivo, we electroporated in utero PHF8-specific shRNA into the developing mouse cortex at embryonic day 14.5 (E14.5), a timepoint at which PHF8 is expressed (Fig. 1A). At this developmental stage, neural progenitors lining the ventricular wall (radial glial cells, RGCs) mainly undergo asymmetric cell division to generate neurons of the upper layer of the cerebral cortex (direct neurogenesis). Cortical neurons are also generated by indirect neurogenesis *via* intermediate progenitors (IPs). Once generated in the ventricular zone

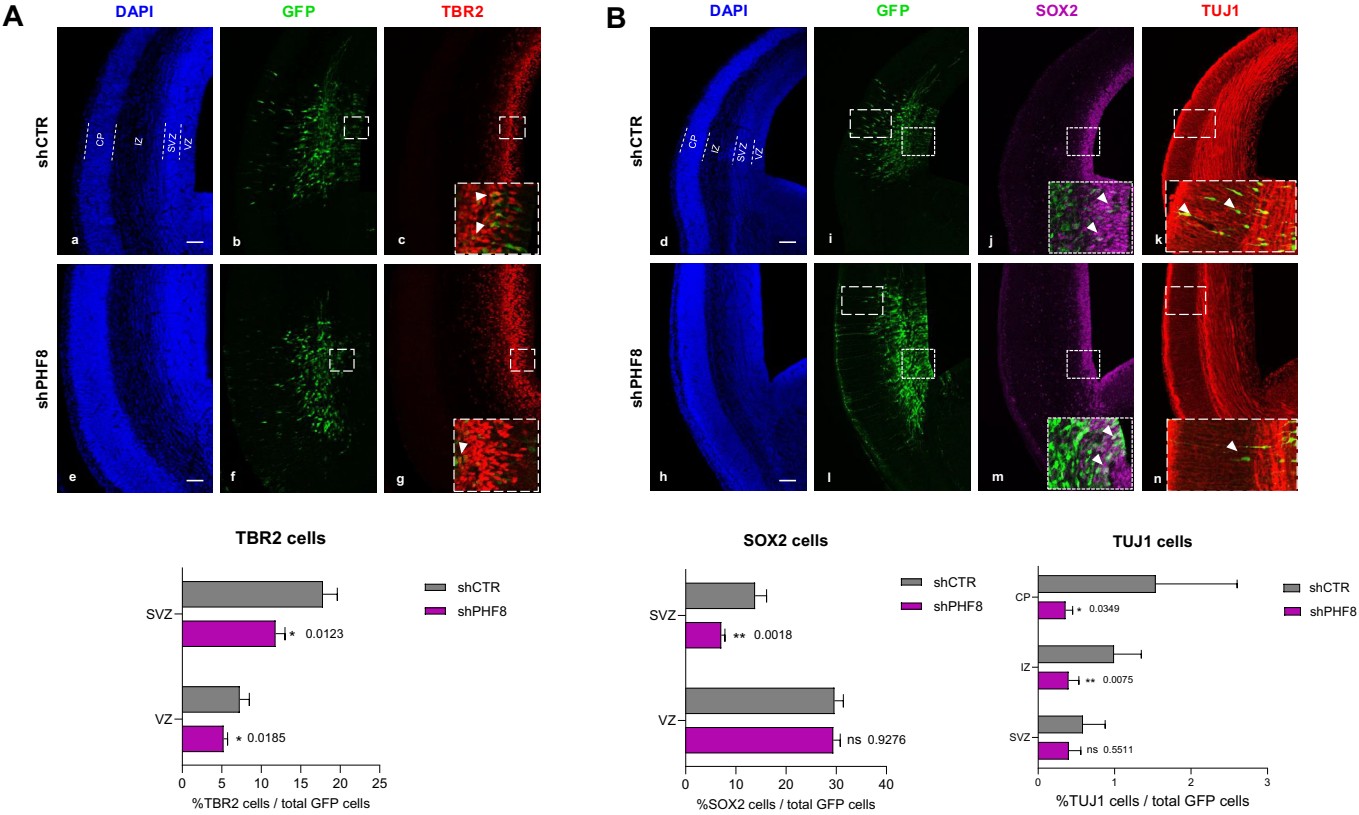

**Figure 6. PHF8 depletion reduces mouse neurogenesis in vivo and impairs the differentiation of the neuronal outputs.**

(**A, B**) E14.5 mouse embryos were electroporated in utero (IUE) with either shCTR or shPHF8, along with GFP-expressing vector. After 48 h post-electroporation (PE), brains were dissected. Transverse sections were fixed and analyzed by immunostaining using antibodies specific to SOX2 (panels c, g) and TUJ1 (panels d, h) in (**A**) and TBR2 (panels k, n) in (**B**); and DAPI (panels a, e, i, l). Electroporated cells were visualized via GFP expression. Scale bar represents 100 μm. Quantification plots show the percentage of GFP-positive cells expressing each marker. Data represent the mean from four to twelve embryos were derived from three shCTR and four shPHF8 biologically independent experiments. Error bars represent mean ± SEM. ns not significant, two-tailed Mann–Whitney test. Scale bar, 100 μm. White arrows indicate GFP+ cells expressing the specific marker (SOX2, TBR2, or TUJ1). CP cortical plate, IZ intermediate zone, SVZ subventricular zone, VZ ventricular zone Source data are available online for this figure.

(VZ), IPs migrate to the subventricular zone (SVZ), downregulate the RGC marker SOX2, and upregulate TBR2—an IP marker—or early neuronal markers such as doublecortin (DCX) and TUJ1. Consequently, impairment of RGC cell cycle progression is expected to reduce the production of IPs and neurons.

We analyzed the dynamics of progenitor populations and the neuronal output of electroporated GFP+ cells 48 h after *in utero* electroporation (IUE) at E16.5. The proportion of GFP+ RGCs remaining in the VZ that continued to express the SOX2 marker was similar between shPHF8 and shCTR conditions (Fig. 6A), indicating that RGC identity and rates of self-renewal were not affected. In contrast, we observed a significant reduction in the proportion of SOX2+ cells in the SVZ (%SOX2+/GFP+ cells: shCTR 13.86 ± 2.27, shPHF8 6.51 ± 0.47; $P < 0.05$; Fig. 6A c, g), which reflected a decrease in IPs. Consistent with this, the number of TBR2+ cells was reduced in both the VZ (%TBR2+/GFP+ cells VZ: shCTR 8.11 ± 1.009, shPHF8 5.24 ± 0.51; $P < 0.05$) and the SVZ (%TBR2+/GFP+ cells: shCTR 17.82 ± 1.82, shPHF8 11.83 ± 1.18; $P < 0.05$; Fig. 6B k, n), suggesting a delay or blockade in differentiation.

Most notably, the proportion of GFP+ cells expressing TUJ1 was dramatically reduced in both the intermediate zone (IZ) and the cortical plate (CP) in the shPHF8 condition compared to controls (% TUJ1+/GFP+IZ cells: shCTR 0,992 ± 0.36, shPHF8 0.4 ± 0.14; $P < 0.05$; %TUJ1+/GFP+CP cells: shCTR 1.54 ± 1.065, shPHF8 0.36± 0.093; $P < 0.05$; Fig. 6A, panels d, h). The reduction in TUJ1 cells indicated reduced neuronal production in shPHF8 condition, in agreement with the slower cell cycle re-entry observed in vitro.

To further investigate neuronal production rates, we administered a single pulse of EdU to the dams 34 h after IUE. To estimate the fraction of neurons generated under each condition, we quantified, at E16.5 (14 h post-EdU injection), the number of electroporated (mCherry+) EdU+ cells within the proliferative zones (VZ and SVZ) and the IZ. EdU+ cells in the VZ–SVZ correspond to dividing progenitors, whereas those in the IZ represent their most recent neuronal output. Quantification revealed that the leaving fraction (portion of cells exiting the cell cycle prior EdU administration) was significantly lower in the shPHF8 condition (Fig. EV3A–C), consistent with slower rates of cell cycle exit in PHF8-depleted progenitors (Fig. EV3A,B).

In a complementary experiment, we electroporated a reporter plasmid for DCX expression—an early marker of neuronal differentiation transiently expressed in migratory cortical neurons (Nacher et al, 2001). This analysis revealed that shPHF8-expressing cells failed to activate DCX expression, indicating a blockade in neuronal differentiation (Fig. EV3D–F). Together, these results suggest that PHF8 depletion does not induce aberrant neurogenesis, but rather impairs or delays the normal neuronal differentiation of the neurogenic output.

Taken together, our in vivo data identify PHF8 as a critical regulator of RGC proliferation and demonstrates its essential role in ensuring proper neuronal output during cortical development.

## Discussion

In this study, we demonstrate that the lysine demethylase PHF8 is essential for the transcriptional activation of the serine biosynthetic pathway. PHF8 is doing that by maintaining chromatin in a transcriptionally competent state at the promoters of key serine metabolic genes. Furthermore, we provide evidence that PHF8 acts as a key regulator linking serine availability, transcriptional regulation of its biosynthesis, and autophagy activation, directly impacting cell cycle progression and proliferation in NSCs, and plays a critical role in cortical neurogenesis. Our findings reveal an epigenetic mechanism governing amino acid metabolism and offer a molecular explanation for the functional and pathological role of PHF8 in neural development (Abidi et al, 2007; Koivisto et al, 2007; Laumonnier et al, 2005; Siderius et al, 1999).

Cell growth and proliferation rely on protein synthesis, a process that must be finely coordinated with nutrient availability, particularly amino acids (Zhu and Thompson, 2019). This regulation is especially critical during development when neural progenitors must expand to generate the appropriate number of nervous system cells. However, the mechanisms linking these cellular processes remain poorly understood. Our study suggests that PHF8 is a key factor in maintaining metabolic homeostasis. PHF8 promotes the transcriptional activation of serine biosynthesis genes under conditions of scarcity. The loss of PHF8 triggers a transcriptional program characterized by the repression of these genes and inhibition of autophagy, leading to decreased serine and other amino acids biosynthesis, DNA damage, and growth defects.

Previous studies have highlighted the role of the histone methyltransferase G9A (Ding et al, 2013), and the HDM KDM4C (Zhao et al, 2016) in activating the SBP by maintaining H3K9me1 levels. Our findings suggest that specific epigenetic modifications, H3K9me1 deposition and H4K20me1 demethylation, could work in coordination to regulate SBP gene expression. Therefore, we propose that PHF8-mediated H4K20me1 demethylation is an important regulatory link between amino acid sensing, autophagy, and cell proliferation in NSCs.

Moreover, we show that PHF8 requires NFIA, and to a lesser extent on ATF4, for the transcriptional activation of the serine biosynthetic pathway and the promotion of cell proliferation. NFIA, a nuclear factor with multiple regulatory functions, is directly regulated by PHF8 (Iacobucci et al, 2021), establishing a feedback loop that controls nutrient synthesis. Member of the NFI family play critical roles in embryogenesis, including neocortical,

hippocampal, retinal, and cerebellar development (Clark et al, 2019; Harris et al, 2016; Piper et al, 2014). They are also essential for axon guidance, neuronal differentiation, and migration (Quist et al, 2022; Sagner et al, 2021; Santo et al, 2023; Tchieu et al, 2019), processes that PHF8 has also been shown to regulate (Asensio-Juan et al, 2012). Additionally, NFIA has been implicated in metabolic control, particularly in energy metabolism (Hiraike et al, 2023). Although the precise mechanistic details remain to be elucidated, our findings suggest that PHF8 and NFIA cooperate in amino acid sensing and metabolic regulation through a feedback loop that controls nutrient synthesis.

Mutations in PHF8 have been linked to Siderius-Hamel syndrome, an X-linked intellectual disorder (Abidi et al, 2007; Koivisto et al, 2007; Laumonnier et al, 2005; Siderius et al, 1999). However, its role in neural development remains incompletely understood. Our results highlight a crucial function of *Phf8* during neurogenesis and provide a molecular explanation by revealing an epigenetic mechanism that regulates amino acid metabolism. Disruption of this mechanism affects cell cycle progression and neural progenitor proliferation, ultimately impacting normal nervous system development. These findings suggest that some of the phenotypes observed in patients carrying *PHF8* mutations may result from metabolic alterations affecting early neural development. Furthermore, they raise the possibility that dietary interventions targeting serine availability could represent a potential therapeutic strategy for these patients.

Although our study highlights the metabolic consequences of PHF8 loss and their impact on proliferation and neurogenesis, we acknowledge that additional regulatory layers likely shape the observed phenotypes. Notably, genes governing cell cycle progression and cytoskeletal organization—previously linked to PHF8 function by our group and others (Asensio-Juan et al, 2012; Liu et al, 2010; Sun et al, 2015)—are consistently deregulated in our transcriptomic analyses. These changes, whether independent of or parallel to metabolic alterations, may substantially influence the final outcome. Thus, while we uncover for the first time a direct contribution of metabolic dysregulation to impaired neurogenesis and proliferation in vivo, this mechanism likely acts in concert with other pathways, reflecting the multifaceted nature of PHF8-dependent phenotypes.

Our work opens new avenues for investigating the contribution of PHF8 to cell proliferation and genomic stability in other cellular contexts. Notably, other highly proliferative cells, as cancer cells, exhibit increased activation of the serine-glycine biosynthetic pathway (Geeraerts et al, 2021; Locasale, 2013; Pan et al, 2021; Possemato et al, 2011; Sun et al, 2023; Yang and Vousden, 2016) to meet their growth demands (DeBerardinis et al, 2008). Given that PHF8 is overexpressed in various cancers, (Cheng et al, 2020; Fan et al, 2024; Liu et al, 2020; Liu et al, 2023; Lv et al, 2017; Shao et al, 2017; Tao et al, 2023; Wu et al, 2024; Ye et al, 2019) our findings suggest that this epigenetic regulator may play a key role in activating this pathway in both neural stem cells and tumor cells. By promoting the diversion of glycolytic carbon into macromolecule biosynthesis, PHF8 contributes to the generation of a metabolic microenvironment that supports cell proliferation.

In this and other contexts, PHF8 emerges as a critical regulator of cell proliferation, survival, and differentiation, with

significant implications for both cancer and neurodevelopmental disorders. Understanding how PHF8 influences cellular metabolism will be instrumental in elucidating the crosstalk between development and disease and may offer novel therapeutic opportunities.

# Methods

## Reagents and tools table

| Reagent/ resource | Reference or source | Identifier or catalog number |
|---|---|---|
| **Experimental models** | | |
| Mouse neural stem cells | Cerebral cortices of C57BL/6J mouse fetal brains (E12.5) | |
| Human HEK 293T | | |
| **Recombinant DNA** | | |
| pCMV-VSVG | IRB (Supek's lab) | |
| pCMV-GAL-POL | IRB (Supek's lab) | |
| pLKO.1-control | IRB | |
| pLKO.1-PHF8-1 | IRB | |
| pLKO.1-PHF8-2 | IRB | |
| pLKO.1-NFIA | IRB | |
| pLKO.1-ATF4 | IRB | |
| pInducer-hPHF8 | Cloned by Claudia Navarro | |
| **Antibodies** | | |
| PHF8 | Abcam | ab36068 |
| H4K20me1 | Abcam | ab9051 |
| H3K9me2 | Abcam | ab1220 |
| H3K9me3 | Abcam | ab8898 |
| α-tubulin | Abcam | ab4074 |
| p-H2A.X (Ser139) clone JBW301 | Merck | 05-636 |
| IdU (BrdU) | BD Biosciences, | Cat n° 34758 |
| CldU (BrdU) | Abcam | ab6326 |
| LC3 | MBL | PM036 |
| SQSTM1 | Progen | GP62-C |
| S6K | Cell Signaling | 9202S |
| p-S6K (Thr389) | Cell Signaling | 9234S |
| β-actin | Merck | A2228 |
| TFEB | ProteinTech | 13372-1-AP |
| GFP | AvesLabs | GFP-1010 #GFP3717982 |
| SOX2 | R&D Systems | AF2088 |
| TBR2 | Abcam | 183991 |
| TUJ1 | Biolegend | 801202 |

| Reagent/ resource | Reference or source | Identifier or catalog number |
|---|---|---|
| **Oligonucleotides and other sequence-based reagents** | | |
| Phf8 | This study. Sense primer: CCGCCCAACAAATGCTAATC Antisense primer: GAGGCAGGGTGTCTTCTATTT | Figs. 1B and 2F, G; Appendix Figs. S1B, C, S2B; Fig. EV2D |
| Gapdh | This study. Sense primer: GGAGAAACCTGCCAAGTATGA Antisense primer: CCTGTTGCTGTAGCCGTATT | |
| Phf2 | This study Sense primer: CCCTGGAGTCTTTCTCACAC Antisense primer: CCGTTCCGATGGATCTTCAAG | Appendix Fig. S1B, C |
| Kiaa17/18 | This study Sense primer: GAGCACAGAGGAAGAAGCTATT Antisense primer: CTCAGGCTGCCATTACTTATCT | Appendix Fig. S1B, C |
| Phgdh | This study Sense primer: GGGCATCCTAGTCATGAACAC Antisense primer: CCCATTTGCCATCTTTCATCG | Figs. 2F, G and EV2D; Appendix Figs. S2B and S5B |
| Psat1 | This study Sense primer: GCTGCCACACTCGGTATTG Antisense primer: AGCTAGCAATTCCCTCACAAG | Figs. 2F, G and EV2D Appendix Figs. S2B and S5B |
| Psph | This study Sense primer: GATAGCACCGTCATCAGAGAAG Antisense primer: ATGAGGAACACCTGGACATTAC | Figs. 2F, G and EV2D Appendix Figs. S2B and S5B |
| Pgk1 | This study Sense primer: GCCAAGTCCGTTGTCCTTAT Antisense primer: TCCCTTCCCTTCTTCCTCTAC | Fig. 2F |
| Nfia | This study Sense primer: CCTCCAACCACATCAACAGAAG Antisense primer: GTACCAGGACTGTGTCTGTTG | Fig. EV2D |
| Kdm1a | This study Sense primer: AACTATGTAGCTGATCTTGGCG Antisense primer: CATTGGCTTCATAAAGTGGGC | Fig. EV2D |
| Glud1 | This study Sense primer: GAACTATTCCTGTGGTCCCC Antisense primer: GTTATACTTCATGGCTGTGCG | Appendix Fig. S2B |
| Gpt2 | This study Sense primer: GGCAGCTCAGTCCCATAAAAT Antisense primer: CTTGGGAGGGTCTGGCTC | Appendix Fig. S2B |
| Idh1 | This study Sense primer: AGTGACAGGCTGGGTAAAAC Antisense primer: CACCTTCTGAGTTCCATCTTTTG | Appendix Fig. S2B |
| Pten | This study Sense primer: CTGCCAGCTAAAGGTGAAGATA Antisense primer: TCCTCTGGTCCTGGTATGAA | Appendix Fig. S2B |
| Mitf | This study Sense primer: AGTGAGTGCCCAGGTATGA Antisense primer: GACAGGAGTTGCTGATGGTAAG | Appendix Fig. S2B |

| Reagent/ resource | Reference or source | Identifier or catalog number |
|---|---|---|
| *Atf4* | This study Sense primer: ACCTTCGAGTTAAGCACATTCC Antisense primer: CTGTTCAGGAAGCTCATCTCG | Appendix Fig. S5B |
| *Phgdh* TSS | This study Sense primer: TGCTAGAGTCAGGCCTTAGA Antisense primer: AAAATGTGTGGGAGGCTCTG | Appendix Fig. S3B |
| *Glud1* TSS | This study Sense primer: GCCGAAGTCCGTCCT Antisense primer: CTGCTTGTCTGGCTGT | Appendix Fig. S3B |
| *E2f4* TSS | This study Sense primer: GGAGTTGCACCAGATACCC Antisense primer: AACTTGGTGGTGAGAAGTCC | Appendix Fig. S3B |
| *Prcc* TSS | This study Sense primer: CTACCTCCGGGAACTCA Antisense primer: CTTTACGCCACCTTTAGACT | Appendix Fig. S3B |
| **Chemicals, enzymes and other reagents** | | |
| Doxycycline | Merck | 324385 |
| Puromycin | Merck | P8833 |
| G418 | Merck | 345810 |
| Concanamycin A | Merck | C9705 |
| CBR-5884 | MedChemExpress | HY-100012 |
| **Software** | | |
| **Other** | | |

## Cell culture

Mouse neural stem cells (NSCs) were isolated from the cerebral cortices of C57BL/6J mouse fetal brains (E12.5) and cultured in dishes pre-coated with poly-D-lysine (5 µg/ml, 2 h at 37 °C) and laminin (5 µg/ml, 6 h at 37 °C), following previously established protocols (Currle et al, 2007). NSCs were maintained in a medium consisting of equal parts of DMEM/ F12 (without Phenol Red, Gibco) and Neurobasal medium (Gibco), supplemented with Penicillin/Streptomycin (5%), Gluta-max (1%), N2 and B27 supplements (Gibco), sodium pyruvate (1 mM), non-essential amino acids (0.1 mM), Heparin (2 mg/l), Hepes (5 mM), bovine serum albumin (25 mg/l), and β-mercaptoethanol (0.01 mM), as described previously (Estaras et al, 2012). Fresh recombinant human epidermal growth factor (EGF) (R&D Systems) and fibroblast growth factor (FGF) (Invitrogen) were added to the medium at final concentrations of 20 ng/ml and 10 ng/ml, respectively. Under these conditions, NSCs maintain their capacity for self-renewal and differentiation into various neural cell types (Currle et al, 2007; Pollard et al, 2006).

Human HEK 293T and HeLa cells were cultured under standard conditions (Sanchez-Molina et al, 2014), in Dulbecco's modified Eagle's medium supplemented with 10% fetal bovine serum and 1% Penicillin/Streptomycin.

Cells were maintained in incubators at 37 °C, with a partial $CO_2$ pressure of 5%. The cell lines were recently authenticated and frequently tested for mycoplasma contamination.

## Mice and ethics

Wild-type (WT) C57BL/6JRccHsd mice (Envigo Laboratories, formerly Harlan, Indianapolis, USA) were used in all experiments. The morning of vaginal plug appearance was designated as embryonic day 0.5 (E 0.5). Animals were housed and maintained following the guidelines from the European Union Council Directive (86/609/European Economic Community). All procedures for handling and sacrificing animals adhered to relevant ethical regulations for animal testing and research. The experiments were performed under the European Commission guidelines (2010/ 63/EU) and were approved by the CSIC and the Community of Madrid Ethics Committees on Animal Experimentation, in compliance with both national legislation and the European Union Council Directive (86/609/European Economic Community).

## In utero electroporation (IUE)

In utero electroporation was performed as previously described (Briz et al, 2017). Two pregnant C57BL/6J (wild-type) mice (Charles River Laboratories) were anesthetized with isoflurane/ oxygen. At embryonic stage E14.5, embryos were injected into the lateral ventricle using a 30-mm pulled glass micropipette, with a plasmid solution (1 µg/µl) containing pCAG-GFP (Addgene #11150) along with either control shRNA (shCTR) or PHF8-targeting shRNA (shPHF8). Five voltage pulses (36 mV, 50 ms) were applied with platinum tweezer-type electrodes (Sonidel Limited, #CUY650P5) oriented to target the somatosensory cortex. Brains were collected 48 h later (at E16.5) and fixed in 4% paraformaldehyde (PFA) for subsequent immunohistochemical analysis.

## EdU injection and analysis

Pregnant dams were intraperitoneally injected with the thymidine analog, 5-Ethynyl-2'-deoxyuridine (EdU) (*Merk* cat. 900584), (50 mg/kg body weight), 34 h after IUE at E14.5, using a 30G needle. Brains were collected at E16.5, sectioned, and processed for EdU detection using the Click-It Alexa Fluor imaging kit (Thermo Fisher Scientific, cat. C-10337 and C-10340), following the manufacturer's instructions. To estimate the rate of cell cycle exit in EdU experiments, we calculated the proportion of EdU-negative cells among electroporated cells. These EdU-negative cells correspond to progenitors that were not in S-phase during the EdU pulse and have therefore diluted the EdU signal through subsequent cell divisions.

## Primary antibodies and reagents

Antibodies used were anti: PHF8 (Abcam, ab36068), H4K20me1 (Abcam, ab9051), H3K9me2 (Abcam, ab1220), H3K9me3 (Abcam, ab8898), NFIA (Abcam, ab 228897), ULK1 (Cell Signaling, #8054), p-ULK1 (ser 757) (Cell Signaling, #6888), alpha TUBULIN (Abcam, ab4074), p-H2A.X (Ser139) clone JBW301

(Merck, 05-636), IdU (BD biosciences, Cat n° 34758), CldU (Abcam, ab6326), LC3 (MBL, PM036), SQSTM1 (Progen, GP62-C), S6K (Cell Signaling, #9202S), p-S6K (thr389) (Cell Signaling, #9234), B-actin (Merck, A2228), TFEB (ProteinTech, 13372-1-AP), GFP (AvesLabs GFP-1010), SOX2 (R&D Systems AF2088), TBR2 (Abcam 183991), TUJ1 (Biolegend 801202), DAPI (Thermo Fisher, D1306), and DCX (Millipore, 324385) was used at 1 µg/ml.

## Plasmids

The previously described lentiviral vectors were either obtained from Sigma or cloned into the pLKO.1 puro vector using AgeI and EcoRI restriction sites, with target sequences indicated in brackets: pLKO-random (CAACAAGATGAAGAGCACC) and pLKO-PHF8 (GCAGGTAAATGGGAGAGGTT), pLKO-PHF8-2 (GCAAGAT-GAAACTCGGTGATT), pLKO-mNFIA (GCGCAGTTACAACTT-CACTAT), and pLKO-ATF4 (CCAGAGCATTCCTTTAGTTTA). The pInducer vector expressing PHF8 WT has been reported previously (Iacobucci et al, 2021) and was induced by the addition of doxycycline (1 µg/ml). The pCMV-GFP plasmid (Addgene #11153) was used as a transfection control. The CAG-mCherry construct was a gift from Jordan Green (Addgene plasmid #108685; http://n2t.net/addgene:108685; RRID:Addgene_108685) (Mishra et al, 2019). The pCAG-GFP plasmid was a gift from Connie Cepko (Addgene plasmid #11150; http://n2t.net/addgene:11150; RRID:Addgene_11150) (Matsuda and Cepko, 2004). The Dcx promoter reporter plasmid (Dcx-mCherry) was kindly provided by Qiang Lu (Wang et al, 2007).

## Lentiviral transduction

Lentiviral transduction was carried out as previously described (Asensio-Juan et al, 2017). In brief, HEK 293T cells were transfected with calcium phosphate and a mixture of shRNA transfer vector DNAs, packaging and envelope plasmids (6, 4.5, and 1.5 µg, respectively). After 24–72 h, the medium was collected, and viral particles were concentrated by ultracentrifugation (26,000 rpm, 2 h, at 4 °C). The concentrated viral particles were then used to transduce NSCs. After 24 h, cells were subjected to selection using puromycin (2 µg/ml) for pLKO.1 vectors and neomycin (600 µg/ml) for pInducer vectors. Following selection, between 99–100% of the cells expressed the shRNA.

## Proliferation assay

Cells were seeded in a 12-well plate and counted in triplicate at 0, 24, 48, and 72 h using the Invitrogen Countess II automated cell counter.

## Cell cycle analysis

Cells were incubated with BrdU for 30 min. Following BrdU labeling, cells were trypsinized and fixed in 70% ethanol overnight at −20 °C. After washing, DNA was denatured with 2 M HCl in PBS containing 0.1% (v/v) Triton X-100 for 70 min. The acid solution was neutralized using 0.1 M sodium borate buffer ($Na_2B_4O_7 \cdot 10H_2O$, pH 8.5). Cells were then blocked with 3% BSA in PBS-T [0.05% (v/v) Tween-20]. Primary antibodies—anti-BrdU (Abcam, ab6326; 1:500) and anti-MPM2 (Mitotic Protein

Monoclonal 2) (Millipore, 05-368; 1:1000)—were incubated for 1 h at room temperature. Secondary antibodies—Rat Alexa Fluor 488-conjugated (1:500), and Mouse Alexa Fluor 647-conjugated (1:1000)—were incubated for an additional 1 h at room temperature. Finally, cells were washed and resuspended in a commercial propidium iodide (PI) solution (BD Pharmingen™ PI/RNase Staining Buffer, 550825). Stained cells were analyzed using a CytoFLEX flow cytometer, and data were processed with CytExpert 2.6 software (Beckman Coulter, Inc.).

## Indirect immunofluorescence and quantification

Immunofluorescence assays were performed as previously described (Sanchez-Molina et al, 2014). Cells were fixed in PFA for 20 min and subsequently permeabilized with methanol for 10 min. Blocking was carried out at room temperature for 1 h using 5% bovine serum albumin (BSA) in PBS containing 0.1% Triton X-100. Primary antibodies were diluted in 5% BSA and incubated with the cells for 1 h at room temperature. After washing, cells were incubated for an additional hour at room temperature with Alexa Fluor-conjugated secondary antibodies (1:10,000) (also diluted in 5% BSA) along with 0.1 ŋg/µl DAPI (Sigma) for nuclear staining. Coverslips were then mounted using ProLong™ Glass Antifade Mountant (Invitrogen). Images were acquired using a Zeiss LSM780 confocal microscope and analyzed with ImageJ software. Two distinct quantification methods were applied depending on the experimental context. For histone modifications and γ-H2AX, fluorescence intensity per cell was quantified using ImageJ software. Corrected cellular fluorescence was calculated using the formula: Integrated density/area of the selected cell. For the analysis of the autophagic marker LC3, LC3-I expression was assessed by calculating the total integrated fluorescence intensity normalized to the area of the selected cell (Integrated density/area). LC3-II was used as an indicator of autophagic vesicles (AVs).

The number of AVs was quantified by counting the fluorescent LC3-II foci per cell, while the size of each autophagic vesicle was determined by measuring the area of individual foci using ImageJ software.

## Mouse embryo cortex indirect immunofluorescence

Brains were fixed overnight in 4% PFA, then post-fixed in 0.4% PFA for 72 h, and finally stored in 1X PBS at 4 °C. Then, brains were embedded in 2% agarose, and 80-µm-thick sections were obtained using a Leica VT1000S vibratome (Leica Biosystems). Sections were stored at –20 °C in cryoprotective solution (1X PBS containing 30% glycerol and 30% ethylene PBS).

For immunohistochemistry, sections were permeabilized in 0.5% Triton X-100 in PBS for 30 min and then blocked for 1 h at room temperature in 10% fetal bovine serum (FBS) in PBS with 0.5% Triton X-100. Primary antibodies were diluted in 5% FBS with 0.5% Triton X-100 and 1:1000 sodium azide, and incubated for 48 h at 4 °C. After washing, sections were incubated for 1 h at room temperature with Alexa Fluor-conjugated secondary antibodies diluted in 0.5% FBS, together with DAPI (0.1 ŋg/µl, Sigma) for nuclear staining. For SOX2 detection, an antibody amplification step was included, involving incubation with a biotinylated secondary antibody prior to fluorophore-conjugated streptavidin staining to enhance signal visualization

## RNA extraction and qPCR

RNA was extracted using TRIZOL reagent (Life Technologies) according to the manufacturer's instructions. Reverse transcription was carried out with 500 ng of RNA using the high-capacity cDNA Reverse Transcription Kit (Life Technologies). qPCR was performed with SYBR Green (Roche) on a QuantStudio5 Real-Time PCR System (Applied Biosystems).

## Protein extraction and Western blot

Total protein extraction was performed using RIPA buffer (50 mM Tris-HCl pH 8, 1 mM EDTA, 0.5 mM EGTA, 1% Triton X-100, 0.5% sodium deoxycholate, 0.1% SDS, 150 mM NaCl, and protease inhibitors). Immunoblotting was carried out following standard SDS-PAGE procedures, followed by transfer to a nitrocellulose membrane (Cytiva, 10600018) in a semi-dry transfer device (Bio-Rad). Membranes were blocked for 1 h at room temperature with 5% BSA in TBS-T (Tris-buffered saline containing 0.1% Tween-20) before overnight incubation at 4 °C with primary antibodies in 5% BSA in TBS-T. After washing, the membranes were incubated for 1 h at room temperature with the corresponding secondary antibodies (Invitrogen: Anti-mouse A16072, Anti-rabbit A16104 and Anti-guinea pig A18769) in 5% BSA in TBS-T. Finally, the bands were detected with the Chemidoc Imaging System (Bio-Rad) using an enhanced chemiluminescence kit (Cytiva, RPN2236). Immunoblot relative quantification was performed using Image Lab software (Bio-Rad).

## ChIP-seq procedure and analysis

ChIP assays were performed as previously described (Fueyo et al, 2018) with modifications. A total of $5 \times 10^6$ NSCs were cross-linked using 0.4% Cross-link Gold (Diagenode, C01019027) in PBS for 30 min, followed by fixation with 1% methanol-free formaldehyde for 10 min. The cross-linking reaction was stopped by incubating the cells with 0.125 mM glycine for 10 min. Subsequently, cells were sequentially lysed using three different lysis buffers: lysis buffer 1 (50 mM HEPES, 140 mM NaCl, 1 mM EDTA, 10% glycerol, 0.5% NP-40, and 0.25% Triton X-100), lysis buffer 2 (10 mM Tris, 200 mM NaCl, 1 mM EDTA, 0.5 mM EGTA), and lysis buffer 3 (10 mM Tris, 100 mM NaCl, 1 mM EDTA, 0.5 mM EGTA, 0.1% sodium deoxycholate, and 0.5% N-lauroylsarcosine). Chromatin was fragmented using a Bioruptor sonicator (Diagenode) prior to immunoprecipitation. Primary antibody (Abcam, ab9051) was used for immunoprecipitation. The resulting chromatin–antibody complexes were captured using magnetic beads (Magna ChIP™ Protein A Magnetic Beads, Millipore, 16-661). Following decrosslinking, DNA was then purified by phenol-chloroform extraction and ethanol precipitation.

Raw data quality control and processing were performed by BGI Genomics. Clean reads were aligned to the mouse reference genome (mm10) using SOAPaligner/SOAP2 (v2.21t), allowing up to two mismatches. SOAP2 is a high-speed aligner optimized for short-read sequences generated by platforms such as the Illumina Genome Analyzer.

Peak calling was performed using SICER2, a tool designed to detect broad enrichment regions typical of histone modifications like H4K20me1. SICER segments the genome into windows, scores read enrichment using a Poisson distribution, and identifies clusters of enriched signal ("islands"), allowing adjustable gaps to account for variability in ChIP-seq data. For genome-wide signal quantification and comparison, the genome was divided into 30 kb non-overlapping bins, and changes in ChIP-seq signal were analyzed by comparing enrichment levels in each bin between conditions, identifying upregulated and downregulated regions accordingly.

The pipeline used to analyze ChIP-seq data has been uploaded to GitHub https://github.com/ClinicalTranslationalBioinformatics/artes_et_al_2025

To visualize ChIP-seq data, we used the Integrative Genomics Viewer (IGV) v2.13.2.

## Assay for transposase-accessible chromatin (ATAC-seq) procedure

ATAC-seq was performed as previously described (Buenrostro et al, 2013) with some modifications. In brief, $3.2 \times 10^6$ cells were scraped and resuspended in RBS buffer (10 mM Tris-HCl, pH 7.4; 10 mM NaCl; 3 mM MgCl$_2$), followed by the addition of 0.1% Igepal CA-630 for cell lysis. After this, 50,000 nuclei underwent a transposition reaction using the Tn5 Transposase enzyme and buffer kits (Illumina, FC-121-1030). Immediately after transposition, DNA was purified using the Qiagen MinElute PCR Purification Kit (Qiagen, 28004). For library construction, the Nextera DNA Library Prep Kit (FC-121-1030) was used, along with unique dual index barcodes for each sample and NEBNext High-Fidelity 2x PCR Master Mix (New England Lab, M0541). To minimize GC and size bias during PCR, amplification was monitored using a side qPCR to terminate the reaction before saturation. Size selection was performed using Ampure XP beads (Beckman Coulter, A63880). Libraries were then pair-end sequenced on the NextSeq2000, generating approximately 50 M reads per sample. Data quality control and processing were conducted using the nf-core and ENCODE ATAC-seq pipeline. Paired-end reads were aligned to the mouse mm10 reference genome using Bowtie2 (v2.3.5.1) (Langmead and Salzberg, 2012) with the "--very-sensitive-local" mode. SAMtools (v1.9) (Li et al, 2009) was used to filter out low-quality reads using flag 1796, remove reads mapped to the mitochondrial chromosome, and discard those with a MAPQ score below 20. The resulting BAM files were sorted, and deepTools (Ramirez et al, 2016) was used to generate counts per million (CPM) normalized signal tracks (bamCoverage --samFlagInclude 64 --normalizeUsing CPM) in bedGraph and bigWig formats.

## RNA-seq procedure

Total RNA was extracted from two biologically independent samples using the High Pure RNA Isolation Kit (Roche), followed by DNase I treatment to eliminate genomic DNA contamination. RNA-seq libraries were prepared using the TruSeq Stranded Total RNA Sample Preparation Kit in combination with the Ribo-Zero Human/Mouse/Rat Kit (Illumina, RS-122-2201/2), following the manufacturer's protocol. In brief, 500 ng of total RNA was subjected to ribosomal RNA depletion, followed by fragmentation for 4.5 min. The remaining steps of library preparation were carried out according to the manufacturer's instructions. Final libraries were assessed using the Agilent DNA 1000 chip to

evaluate concentration and fragment size distribution. Quantification was performed by qPCR using the KAPA Library Quantification Kit (Roche, 07960204001), prior to cluster generation with the Illumina cBot system. The qualified libraries were sequenced pair end on the BGISEQ-500/ MGISEQ-2000 System (BGI-Shenzhen, China).

## RNA-seq data analysis

RNA-seq analysis was performed by BGI-Shenzhen. Clean reads were mapped to the reference transcriptome using Bowtie2 v2.2.5 with the parameters -q --sensitive --dpad 0 --gbar 99999999 --mp 1,1 --np 1 --score-min L,0,-0.1 -p 16 -k 200 (Langmead and Salzberg, 2012). Gene expression levels were quantified using RSEM v1.2.8 (-p 8 --forward-prob 0 --paired-end) (Li and Dewey, 2011). Differential gene expression analysis was performed with DESeq2 using the negative binomial distribution model, and genes with an adjusted $p$ value ($q$ value) ≤0.05 were considered significant (Love et al, 2014). Hierarchical clustering of differentially expressed genes was performed using the R package pheatmap (default parameters). All analyses were performed with default settings unless otherwise specified.

## Metabolomics

For each experimental condition, $1.8 \times 10^6$ cells were processed in triplicate for metabolite extraction. To isolate intracellular metabolites, cells were first rinsed with ice-cold 150 mM ammonium acetate (pH 7.4), followed by quenching in 1 mL of cold 80% methanol in water. An internal standard, 1 nmol of D/L-norvaline, was added to each sample. The cell suspensions were incubated at −80 °C for 60 min to ensure effective quenching and protein precipitation. Cells were then scraped, transferred to Eppendorf tubes, and vortexed for 10 s. The samples were centrifuged at 16,000×$g$ for 15 min at 4 °C. Supernatants were collected, transferred to glass vials, and dried under vacuum. Metabolite analysis and quantification were performed by the UCLA Metabolomics Center (USA). Dried metabolites were reconstituted in 100 μL 50% (v/v) acetonitrile (ACN) and dH$_2$O solution. After centrifugation for 10 min at 17,000×$g$, 70 μL of supernatant was transferred to HPLC glass vials, and 10 μL of these metabolite solutions were injected per analysis. Metabolites were separated by liquid chromatography (Vanquish, Thermo Fisher Scientific, Waltham, MA, USA) and coupled to a mass spectrometer (Q-Exactive, Thermo Fisher Scientific, Waltham, MA, USA) to obtain ion chromatograms. Peaks were aligned among all samples and assigned identities using exact mass and retention time based on our in-house database. Peaks were quantified by area under the curve integration and normalized by the measured area of the internal standard norvaline. Metabolite abundance was normalized per μg of protein content per metabolite extraction.

## DNA fiber assay

NSCs were sequentially pulse-labeled with CldU and IdU to monitor DNA replication dynamics. After harvesting, DNA fiber spreads were prepared following established protocols (Henry-

Mowatt et al, 2003). CldU and IdU incorporation were detected by sequential immunostaining using anti-BrdU antibodies specific for each analog, followed by Alexa Fluor-conjugated secondary antibodies. Fibers were imaged using a Leica AF6000 microscope with a 40x objective. ImageJ software was used for the quantification of replication structures. Restarted forks (fibers labeled with both CldU and IdU), stalled forks (CldU only), and new origins (IdU only) were counted. Additionally, the length of IdU-labeled tracks (visualized with AlexaFluor 488) was measured.

## Gene set overlap analysis

To assess the likelihood of observing an overlap of 614 genes between the 2946 genes downregulated in the PHF8 RNA-seq dataset and the 2408 genes downregulated in the NFIA RNA-seq dataset by chance, we generated a random distribution of overlap values. This was done by comparing the 2946 gene list with 10,000 random samples of 2408 genes each. Specifically, we drew 10,000 random samples from the 21,810 mouse genes annotated in Ensembl (GRCm39), computed the overlaps, and constructed the resulting empirical distribution (Fig. EV2E).

## Statistical analysis

Quantitative data were expressed as mean ± standard deviation (SD) and as standard error of the mean (SEM). Differences between the two groups were analyzed using the unpaired Student's $t$-test. For nonparametric data, the Wilcoxon signed-rank test or the Mann–Whitney $U$-test was used, depending on the requirements of the analysis. For comparisons involving three or more groups, two-way ANOVA was applied. Asterisks indicate the corresponding p-values, which were calculated using GraphPad Prism 8. The experiments were conducted blindly.

# Data availability

All sequenced data have been deposited in the GEO database under the super series GSE296528. ChIP-seq data have been deposited in the GEO database under the accession GSM8972159, GSM8972160, GSM8972161, GSM8972162, GSM8972163, GSM8972164. RNA-seq data have been deposited in the GEO database under the accession GSM9329339, GSM9329340, GSM9329341, GSM9329342, GSM9329343, GSM9329344. ATAC-seq data have been deposited in the GEO database under the accession GSM8972165, GSM8972166, GSM8972167. H3K9me3 ChIP-seq data have been deposited in the GEO database under the super series GSE311637. accession GSM9328548, GSM9328549, GSM9328550, GSM9328551, GSM9328552, GSM9328553.

# Peer review information

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

## Acknowledgements

We would like to thank Dr. A. Jordan for the reagents; Drs. S. Sánchez-Molina, M. Liesa, E. Mocholi, and A. Vaquero for reagents and advice. Drs. N. Serna-Pujol and Elia Marcos for advice and technical help. Team members for helpful comments and suggestions. This study was supported by grants PGC2018-096082-B-I00 and PID2021-125862NB-I00 from the Spanish Ministry of Economy and 2021AEP079 from the CSIC to MAMB. PID2020-118768RJ-I00; AEI/10.13039/501100011033, a Ramon y Cajal fellowship (RYC2022-035576-I) and LABAE222994MAUV from AECC to CM. PID2022-138728OB-I00 from MICIU/AEI/10.13039/501100011033 and FEDER (UE) to NA. APV was supported by PREDOC-UB 2022 from the University of Barcelona, co-funded by Banco Santander.

## Author contributions

**Marta H Artes**: Conceptualization; Formal analysis; Investigation. **Simona Iacobucci**: Conceptualization; Formal analysis; Investigation. **María J Barallobre**: Conceptualization; Investigation; Methodology. **Paula Carballeira**: Formal analysis; Investigation. **Marta Garcia-Cajide**: Formal analysis; Investigation. **Alejandro Pérez-Venteo**: Formal analysis; Investigation. **Natalia Padilla**: Data curation; Software; Formal analysis; Investigation; Visualization. **Bárbara S Viegas**: Formal analysis; Investigation. **Aitana Díaz-Vásquez**: Formal analysis; Investigation. **A Silvina Nacht**: Investigation. **Guillermo P Vicent**: Formal analysis; Supervision. **Maria L Arbonés**: Conceptualization; Supervision; Writing—review and editing. **Xavier de la Cruz**: Resources; Data curation; Software; Supervision; Writing—review and editing. **Marta Nieto**: Conceptualization; Formal analysis; Supervision; Methodology; Writing—review and editing. **Neus Agell**: Conceptualization; Supervision; Methodology; Writing—review and editing. **Caroline Mauvezin**: Conceptualization; Supervision; Writing—review and editing. **Marian A Martínez-Balbás**: Conceptualization; Formal analysis; Supervision; Investigation; Writing—original draft; Project administration; Writing—review and editing.

Source data underlying figure panels in this paper may have individual authorship assigned. Where available, figure panel/source data authorship is listed in the following database record: biostudies:S-SCDT-10_1038-S44319-026-00713-8.

## Disclosure and competing interests statement

The authors declare no competing interests.

# Expanded View Figures

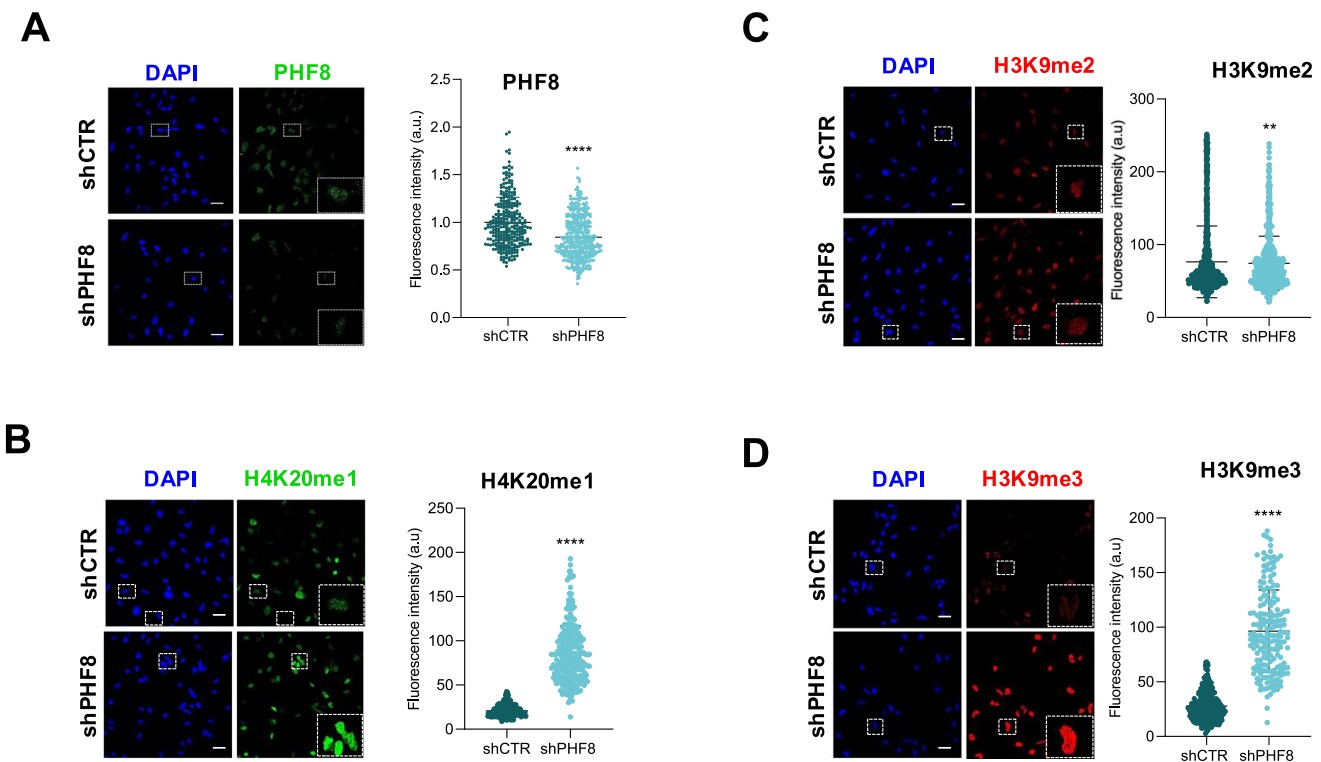

**Figure EV1.  PHF8 maintains transcriptionally competent chromatin.**

(**A–D**) shCTR and shPHF8 NSCs were immunostained using PHF8 (**A**), H4K20me1 (**B**), H3K9me2 (**C**), and H3K9me3 (**D**) antibodies and DAPI. Violin plots show quantification of the fluorescence intensity normalized to cell area ($n \geq 180$). The data shown are representative of three biologically independent experiments. Scale bar indicates 20 µm. Error bars represent mean ± SEM. **$p = 0.0036$; ****$p < 0.0001$, two-tailed $t$-test. Source data are available online for this figure.

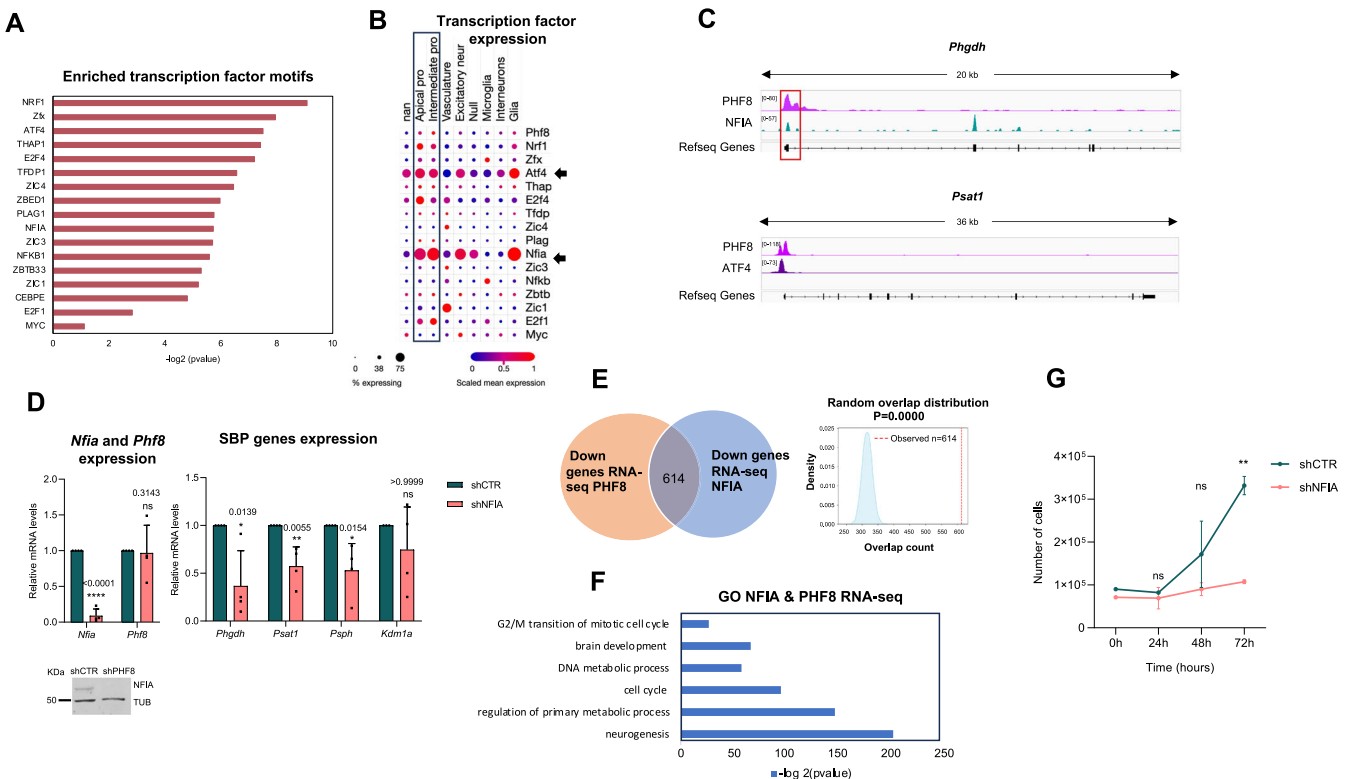

Figure EV2.  PHF8 cooperates with transcription factors to regulate SBP gene transcription.

(A) Motif enrichment analysis of promoters from metabolism-related, PHF8-regulated genes was performed using the PSCAN tool, the top enriched transcription factor motifs. Z-test implemented in PSCAN. (B) Graph showing the expression of *Phf8* and the transcription factors identified in (A) in the developing mouse cerebral cortex. Data are derived from publicly available single-cell RNA-seq datasets (link to dataset). (C) IGV snapshots showing PHF8 binding peaks, NFIA and ATF4 peaks at *Phgdh* (NFIA) and *Psat1* (ATF4) gene promoters. Tracks show the input-subtracted signal. (D) NSCs were infected with lentiviruses expressing shCTR or NFIA-targeting shRNA (shNFIA). Total RNA and protein extracts were collected to assess NFIA protein levels via immunoblotting and mRNA levels of *Nfia*, *Phf8*, *Phgdh*, *Psat1*, and *Psph* by qPCR. Expression levels from four biologically independent experiments were normalized to *Gapdh*, and data were presented relative to shCTR samples. *Kdm1a* was included as a negative control. Error bars represent mean ± SEM. Two-tailed *t*-test was applied. (E) Venn diagram showing the overlap between genes downregulated in our PHF8 KD RNA-seq (Fig. 2) (orange) and genes downregulated upon NFIA knockdown identified in the published NFIA KD RNA-seq dataset (Appendix Table S1) (blue). *p*-value < 0.05. Graph showing the median differences from the permutation test between PHF8 downregulated genes in the RNA-seq dataset and 2048 randomly selected genes (corresponding to the number of genes downregulated in NFIA KD). A total of *n* = 10,000 permutations were performed. The observed overlap value (614) (red vertical bar) was not reached in any of the random samples, resulting in an empirical *p* value of zero (right panel). (F) GO analysis highlighting enriched Biological Processes among genes co-regulated by PHF8 and NFIA identified in (E). *p* value <0.05. Enrichment was assessed by Fisher's exact test, with *P* values corrected using the g:SCS method in g:Profiler. (G) Growth curve showing proliferation rates of NSCs infected with shCTR or shNFIA lentiviruses over 72 h. Data represent the mean of three biologically independent experiments performed in triplicate. Error bars represent SD. **p* = 0.0040; ns not significant, two-tailed *t*-test. Source data are available online for this figure.

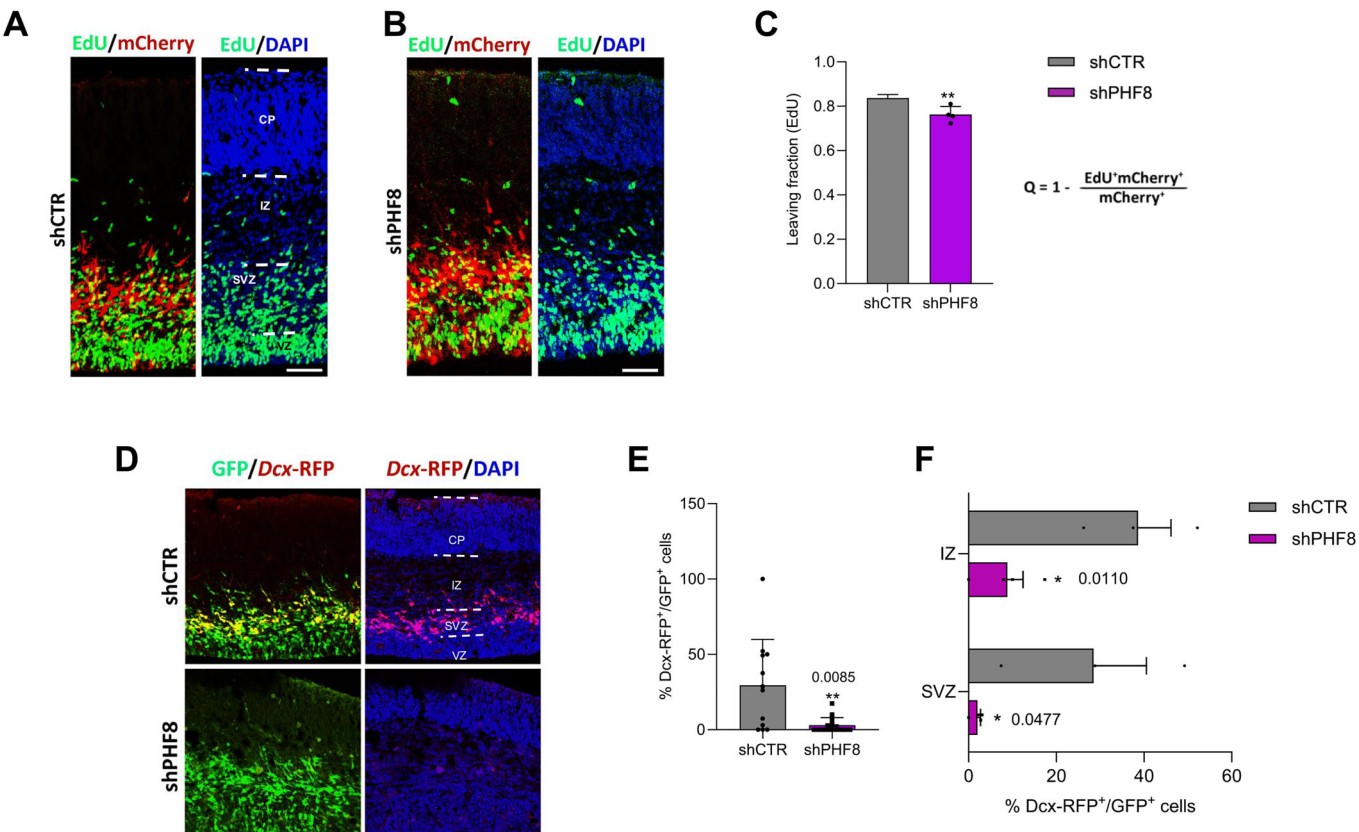

**Figure EV3. PHF8 depletion reduces mouse neurogenesis in vivo and impairs the differentiation of the neuronal outputs.**

(A, B) Representative images of E16.5 brain sections from embryos electroporated in utero at E14.5 with either shCTR or shPHF8 constructs, together with an mCherry reporter. EdU was administered to pregnant dams 34 h post-IUE, and embryos were collected 14 h later for brain dissection. Data represent the mean from four to twelve embryos (derived from at least three shCTR and four shPHF8 biologically independent experiments). Scale bar: 50 μm. CP cortical plate, IZ intermediate zone, SVZ subventricular zone, VZ ventricular zone. (C) Quantification of the global leaving fraction (Q), estimating cell-cycle exit over the 48 h period. Q was significantly reduced in shPHF8-electroporated embryos compared with shCTR. EdU⁺ cells and double EdU⁺mCherry⁺ cells were counted in the VZ–SVZ and IZ of shCTR and shPHF8-electroporated embryos. Data represent mean ± SEM from $n \geq 3$ shCTR and $n = 4$ shPHF8 biologically independent experiments. **$p = 0.0096$, two-tailed $t$-test. (D) Analysis of the early neuronal marker DCX. E14.5 embryos were electroporated with shCTR or shPHF8 together with a GFP-expressing plasmid and a *Dcx* promoter-driven DsRed reporter construct. Data represent the mean from four to twelve embryos (derived from at least three shCTR and four shPHF8 biologically independent experiments). Scale bar: 100 μm. (E, F) Brains were dissected 48 h after electroporation. Quantification of RFP⁺/GFP⁺ cells, shown as total counts (E) and by cortical region (SVZ and IZ) (F), revealed a significant reduction in *Dcx*-RFP reporter expression in the shPHF8 condition compared with shCTR. Data represent mean ± SEM from $n \geq 3$ shCTR and $n = 4$ shPHF8 biologically independent experiments. Two-tailed $t$-test was applied. Source data are available online for this figure.

