## [Peer Review File · EMBO Reports]

Epigenetic Regulation of Serine Biosynthesis by PHF8 During Neurogenesis

Marta Artes, Simona Iacobucci, María José Barallobre, Paula Carballeira, Marta Garcia-Cajide, Alejandro Perez-Venteo, Natalia Padilla, Bárbara Viegas, Aitana Díaz-Vásquez, A. Silvina Nacht, Guillermo Vicent, Maria Arbones, Xavier de la Cruz, Marta Nieto, Neus Agell, Caroline Mauvezin, and Marian Martínez-Balbas

Corresponding author(s): Marian Martínez-Balbas (mmbbmc@ibmb.csic.es)

Review Timeline:

Submission Date:	13th May 25
Editorial Decision:	12th Jun 25
Revision Received:	12th Nov 25
Editorial Decision:	4th Dec 25
Revision Received:	19th Dec 25
Accepted:	15th Jan 26

Editor: Achim Breiling

Transaction Report:

Dear Dr. Martínez-Balbas,

Thank you for the submission of your manuscript to EMBO reports. I have now received the reports from the three referees that were asked to evaluate your study, which can be found at the end of this email.

As you will see, the referees think that these findings are of interest. However, they have several comments, concerns, and suggestions, indicating that a major revision of the manuscript is necessary to allow publication of the study in EMBO reports. As the reports are below, and all the referee concerns need to be addressed, I will not detail them here.

Given the constructive referee comments, I would like to invite you to revise your manuscript with the understanding that the concerns of the referees must be addressed in the revised manuscript and/or in a detailed point-by-point response. Acceptance of your manuscript will depend on a positive outcome of a second round of review. It is EMBO reports policy to allow a single round of revision only and acceptance of the manuscript will therefore depend on the completeness of your responses included in the next, final version of the manuscript.

- 1) a .docx formatted version of the final manuscript text (including legends for main figures, EV figures and tables), but without the figures included. Figure legends should be compiled at the end of the manuscript text.
- 2) individual production quality figure files as .eps, .tif, .jpg (one file per figure), of main figures and EV figures. Please upload these as separate, individual files upon re-submission.

- 4) a complete author checklist, which you can download from our author guidelines (<https://www.embopress.org/page/journal/14693178/authorguide>). Please insert page numbers in the checklist to indicate where the requested information can be found in the manuscript. The completed author checklist will also be part of the RPF.

- 5) that primary datasets produced in this study (e.g. RNA-seq, ChIP-seq, structural and array data) are deposited in an

appropriate public database. If no primary datasets have been deposited, please also state this in a dedicated section (e.g. 'No primary datasets have been generated and deposited'), see below.

The accession numbers and database should be listed in a formal "Data Availability" section that follows the model below. This is now mandatory (like the COI statement). Please note that the Data Availability Section is restricted to new primary data that are part of this study. This section is mandatory. As indicated above, if no primary datasets have been deposited, please state this in this section

Data availability

6) We now request the publication of original source data with the aim of making primary data more accessible and transparent to the reader. You will receive a separate email with instructions for providing source data with your revised manuscript, including information how to upload and organize the files.

8) Regarding data quantification and statistics, please make sure that the number "n" for how many independent experiments were performed, their nature (biological versus technical replicates), the bars and error bars (e.g. SEM, SD) and the test used to calculate p-values is indicated in the respective figure legends (also for EV and Appendix figures). Please also check that all the p-values are explained in the legend, and that these fit to those shown in the figure. Please provide statistical testing where applicable. Please avoid the phrase 'independent experiment', but clearly state if these were biological or technical replicates. Please also indicate (e.g. with n.s.) if testing was performed, but the differences are not significant. In case n=2, please show the data as separate datapoints without error bars and statistics. See also: <http://www.embopress.org/page/journal/14693178/authorguide#statisticalanalysis>

9) Please add scale bars of similar style and thickness to microscopic images, using clearly visible black or white bars (depending on the background). Please place these in the lower right corner of the images themselves. Please do not write on or near the bars in the image but define the size in the respective figure legend.

10) Please also note our reference format:

12) We now use CRedit to specify the contributions of each author in the journal submission system. CRedit replaces the author contribution section. Please use the free text box to provide more detailed descriptions and do NOT provide your final manuscript text file with an author contributions section. See also our guide to authors: <https://www.embopress.org/page/journal/14693178/authorguide#authorshippinguidelines>

13) All Materials and Methods need to be described in the main text using our 'Structured Methods' format, which is required for

all research articles. According to this format, the Methods section should include a Reagents and Tools Table (listing key reagents, experimental models, software, and relevant equipment and including their sources and relevant identifiers), uploaded as separate file, and a Methods section in which we encourage the authors to describe their methods using a step-by-step protocol format with bullet points, to facilitate the adoption of the methodologies across labs. More information on how to adhere to this format as well as downloadable templates (.doc) for the Reagents and Tools Table can be found in our author guidelines (section 'Structured Methods'):

14) Please add up to five keywords to the manuscript and order the sections like this, using these names: Title page - Abstract - Keywords - Introduction - Results - Discussion - Methods - Data availability section - Acknowledgements - Disclosure and Competing Interests Statement - References - Figure legends - Expanded View Figure legends

15) Please make sure that all the funding information is also entered into the online submission system and that it is complete and similar to the one in the acknowledgement section of the manuscript text file.

I look forward to seeing a revised form of your manuscript when it is ready.

Yours sincerely,

Referee #1:

[As a disclaimer: I am an expert in metabolism but not chromatin biology/epigenetics. Thus, I mainly reviewed the manuscript from my perspective of metabolism.]

The study by Artes and colleagues describes an epigenetic regulation of serine biosynthesis through the histone demethylase PHF8. Loss of PHF8 results in lower gene expression of serine synthesis pathway (SSP) genes and subsequently lower intracellular serine and glycine levels affecting nucleotide levels ultimately leading to DNA damage. These observations have implication in neurodevelopment.

Overall the study is nicely done, I only have one concern related to the causality between NFIA and PHF8 as described below:

The conclusion that PHF8 acts through NFIA is not robust. The two factors are not causally linked. The authors silenced both genes independently and only show that loss of either of the two genes affects SSP but this could also be parallel events instead of one common mechanism. For example, I expect that loss of ATF4 would have a similar (if not stronger) effect. In fact, as shown in figure 4A, NFIA only ranks in the middle, while NRF1, Zfx and ATF4 are the top three hits.

The authors try to put NFIA and PHF8 in causal relation by showing a 26.3% overlap in altered gene expression. However, what does this mean? Can the authors benchmark this? How much % could one expect from a common pathway - or how much overlap would one expect when silencing two unrelated genes? In light of the above mentioned, it would be interesting to compare with another known transcription factor (such as ATF4) and/or some factor that is unrelated to PHF8/SSP.

Additionally, the findings on mTORC1 strongly suggest that ATF4 is likely to be involved in the phenotype, since mTORC1 activates the ATF4 axis, which in turn controls amino acid metabolism including serine biosynthesis and nucleotide metabolism.

Minor comment:

In figure 1G, I would express all data in relation to shCtrl +Ser to show also the effect on SSP genes in shPHF8+ser vs shCtrl+Ser. On similar lines, I would merge the two panels in figure 1H as it would also show the lower proliferation rate in shPHF8 at baseline.

Referee #2:

This manuscript by Artes Marta H et al. demonstrated that in neural stem cell (NSC) culture, Phf8 supports NSC proliferation by positively regulating serine biosynthesis. Mechanistically, Phf8, potentially in collaboration with the transcription factor Nf1a, activates transcription of serine biosynthesis pathway genes by demethylating H4K20me1 at their promoters. In addition, Phf8 depletion leads to increased γ H2AX, a critical DNA double strand break biomarker, and impairs autophagy. Using in utero electroporation, the authors also showed that Phf8 loss-of-function reduces asymmetric cell division of neural progenitors in the ventricular zone of mice.

Combining gene manipulation, flow cytometry, multiomics, and other molecular biology and histology methods, the authors clearly characterized Phf8 as an important epigenetic regulator of NSC proliferation and explored the underlying mechanisms. Nevertheless, I believe some evidence needs additional verification or clarification to strengthen the conclusion. Please see below for detailed comments.

Major:

1. The Phf8 shRNA is a key tool in this study. However, almost no experiments were done to exclude the off-target effects of this shRNA. I suggest the authors repeat the NSC proliferation experiments (Figure 1C) with an shRNA targeting another sequence or test if the reduced proliferation could be rescued by overexpressing an shRNA-resistant Phf8 vector.
2. Does Phf8 gain-of-function increase NSC proliferation?
3. No evidence for knockdown of protein was shown for Nf1a shRNA. What is the relationship between Phf8 and Nf1a? Would double knockdown additionally reduce NSC proliferation? Would the reduced NSC proliferation induced by knockdown of Phf8 or Nf1a be reversed by overexpression of the other one?
4. As related to 3, why were Nrf1, Afx, Atf4, E2f4, etc. not investigated for their roles in coordinating with Phf8 to regulate SBP gene expression and NSC proliferation considering their motif enrichment is higher than Nf1a? The authors' statement that they have previously demonstrated Phf8 regulates Nf1a expression does not justify this scientifically.
5. The authors stated that 65% of genes that changed the accessibility upon Phf8 depletion also suffered changes in transcription in the RNA-seq (Figure 3E). First, how was 65% calculated? Second, are changes in chromatin accessibility and gene expression in the same direction? If not, this could be easily interpreted in the opposite way. The second question also applies to Figure 4D.
6. Several previously published sequencing datasets used for analysis in this study were not obtained from NSCs, but were used by the authors to directly compare with their NSC data, which raises the concern of tissue specificity or cell type specificity. This should at least be discussed.
7. In Figure 6A, the representative blot does not seem to represent the quantification result well.

Minor:

1. The source of the single-cell RNA-seq that was used for analysis in Figure 1A was not stated in the text or the appendix table.
2. Please explain in the manuscript how Phf2 and Kiaa1718 are related to Phf8.
3. Figure S1F, please clearly state what data were used for analysis.
4. Figure 2G, please make sure the comparisons are correct.
5. Figure 2H, does NEAA mean no NEAA?
6. Why was $p < 0.08$ instead of 0.05 used as the cut off in RNA-seq analysis?
7. The authors declared there is a global increase in bulk methylation following Phf8 knockdown, however this does not seem to be the case for H3K9me2.
8. Why was a decrease on H4K20me1 levels at the SBP gene bodies upon Phf8 depletion expected due to downregulation of these genes? Please explain and include citations as needed.
9. Methods-Plasmids and recombinant proteins, in what experiment(s) was the pInducer vector expressing wild type Phf8 used?

10. Methods, should the transposase be Tn5 instead of Th5?

Referee #3:

The manuscript of Artes et al. entitled "Epigenetic Regulation of Serine Biosynthesis by PHF8 During Neurogenesis" reports on the role of Phf8 in orchestrating neurodevelopment by adapting metabolic processes, especially the serine metabolism. The main finding described in the manuscript is that Phf8 reduction, mediated through shRNA, impairs proliferation of neural progenitors, which comes together with an altered metabolism, i.e. reduced production of serine, and with increased signs of autophagy. The reported work extends significantly knowledge about phenotypic features to a cell biological and molecular level, and thus provides insight into potential alterations causing a human neurodevelopmental disease. It might be of relevance for a broader community beyond the neurosciences, as Phf8 is also implicated in cancers, and communalities between pathologies might cross-foster the respective other field. Despite that the data are supporting the general line of research presented in the manuscript, I am critical about the robustness of the data, as they are in part not fulfilling all criteria of vigorous experimentation.

The authors claim in their introduction to provide a novel mechanism of integrating cell renewal and metabolic demands. However, in my understanding, this is an overinterpretation of the results. The authors correlate known mechanisms to describe a set of novel targets and metabolic regulation through Phf8. The mechanism that link serine metabolism to either cell cycle progression, or DNA damage, or autophagy that feed into impaired neurogenesis, through the specific function of Phf8 (which has apparently many target genes genome-wide) is not resolved. There might be different layers towards the phenotypic observations, and the authors might take this into account by judging of whether an entire novel mechanism is provided. I address this point, as a weakness of the manuscript is the - in part - insufficient data source that is used for drawing conclusions and interpretations.

Following up on this, some of the experiments do not have the number of replicates necessary to provide statistical analysis, especially the RNAseq data, but also other data are shown with numbers of replicates (two biological replicates) that do not allow a valid statistical testing. This is a real concern as the data might be in part overinterpreted, despite that the overall observed effects might be true. I suggest to provide all data with the appropriate number of replicates (for seq please refer to ENCODE for the present state-of-the-art) and to show all bar graphs with individual data points, to make clear the distribution of the individual observations, which will allow assessing better robust observation from -maybe- random results.

Further comments that the authors might want to address to improve their manuscript:

1. "...processes that occur in parallel with extensive epigenomic remodeling driven by histone acetyltransferases (KATs), histone methyltransferases (KMTs), and histone demethylases (KDMs). Among the former, PHF8, ..." - it is probably more the latter than the former
2. Fig. 1A: Expression over developmental time points is interesting, but should be enriched by untangling the specific cell types, and ordered in pseudotime to display the developmental dynamics in Phf8 expression in cellular distribution. From such an analysis a rationale to focus on NSCs can be derived by the authors, as a rationale is as of yet not given. If Phf8 is also expressed in neurons, it might be interesting to study exemplarily its role in neurons, as these cells switch their metabolism substantially compared to NSCs. If Phf8 specifically impacts amino acid metabolism in NSCs, but not in neurons, these cell-specific functions in regulation metabolism will spice up the findings of the authors.
3. The KD efficiency of Phf8 is around 50%. The authors might want to bring this in the context of whether this model has relevance to the human disease. As shRNAs can have variable and off-side effects, I recommend to use a genetic deletion as additional model to substantiate the findings. In their earlier publication (Iacobucci et al 2021) a similar approach has been used to study Phf8 in astrocytes; to advance towards a gain of specificity using for example a CRISPR/Cas9 approach to generate a genetic model might be of advantage.
4. The cell cycle analysis requires more substance in order to be interpretable - as of yet, the interpretations of the authors are based on non-significant statistical results. I suggest to increase the number of replicates and to provide experimental data on differences in cells leaving or re-initiating the cell cycle. What is the reason for a potential delay in S-phase entry, if this is indeed the main difference upon Phf8 KD? Taking the later finding into account that the replication fork stalls upon Phf8 KD - would this not rather result in S-phase progression defects? Here, clarification is needed benefitting the coherence of the data presented.
5. It is not sufficient to list the packages / tools used for bioinformatics analysis, but it is necessary to provide the versions and the analysis settings in order to adhere to basic principles allowing to reproduce the data. I am missing an accessible repository with all bioinformatics workflows that will allow reproducing the analyses.
6. Along the line of comment #4, it is surprising that DESeq2 tools can handle 1 ctrl and 2 shRNA KD samples to return data of significance, and that the authors can provide a p-value. The RNA-seq experiment is unacceptable, as it is not based on an appropriate numbers of biological replicates for both conditions that are compared. Following ENCODE state-of-the-art, the authors need to improve this experiment and bring it to current standards.
7. The experimental setup of the rescue experiment in Fig. 2F is not given. The degree of OE in the conditions of presence/absence of shRNA should be assessed by immunoblots of PHF8, to provide the levels of PHF8 present in the cell. Seemingly, the OE results in a higher transcriptional response compared to control levels. In addition, the authors should provide a scheme in the supplement illustrating the rescue strategy, where the shRNA is targeting and whether this sequence is

missing in the OE plasmid.

8. Fig. 2G might be moved to a supplement to illustrate that HeLa cells respond to serine-depletion. However, the authors should indicate precisely what has been compared to what in their statistics - each gene has to be compared to its counterpart; the way of representation in its present form is misleading - it might suggest that all conditions might be statistical significant, which I doubt is the case here. As the authors used the Student's t-test, some more parameters should be given (one- or two-sided?) and the authors should test the normal distribution of their data.

9. A new experiment analogous to Fig. 2G using NSCs should be provided. Here, the levels of the control experiment should be compared to the shPhf8 condition as well. And the serine condition should be tested as further variable to discriminate between shPhf8 effects and serine effects.

10. Fig. 2H should be combined to illustrate both conditions together and to give indication of the robustness of shPhf8 KD effect compared to control. The accompanying Fig. S1E shows a milder effect, and not as the authors claim a similar effect, in terms of statistical significance and number of cells. Does the proliferation cease completely if a combination treatment of shPhf8 and PHGDH inhibition is applied?

11. Fig. S2A needs to include expression of PHF8. And levels of the H4K20me1 and H3K9me2 need to be shown in immunoblots together with the respective quantification.

12. Fig. 3B: the authors should provide a k-means clustering with a heatmap illustrating peak enrichment genome-wide, in ctrl and KD conditions.

13. Describing the results of Fig. 3B the authors refer to GO analysis of the regions - GO terms rely on genes that are usually referred to as the ones lying nearest to the peak region. Please describe more exactly.

14. "Among these, the SBP genes (Fig. 3C and Appendix Fig.S2C)." This is not a full sentence.

15. The authors claim that H4K20me1 is the main target of Phf8 in NSCs, based on immunostainings as provided in Fig. S2A. As mentioned above, this finding should be backed up with immunoblots as quantitative method. However, the authors claim to see increase of H4K20me1 in promoter regions, alongside decrease in gene bodies. Thus, the global view point as provided for H3K9me2 might hide changes that go in opposite directions at different locations in the genome as well. Therefore, the interpretation of a main and a minor target might be premature, unless the authors provide clear experimental evidence. H3K9me2 ChIP-seq would be an appropriate means.

16. " This assays assesses" grammar incorrect

17. Fig. 3E: the authors state that 65% of genes with differentially accessible regions are found among differentially expressed genes. However, the numbers given in the figure do not match 65%. The authors should clarify this and provide the numbers underlying the 65% they claim to see, while describing the results in the text.

18. " Thus, we evaluate the impact of NFIA depletion" inconsistent grammar

19. " GO of these genes revealed multiple categories" unprecise phrasing

20. "similar or even higher to the observed upon PHF8 depletion (Fig. 4F)." incomplete sentence

21. Fig. 5B: both conditions of shRNA treatments should be represented in one figure to assess whether serine supplementation would reach ctrl levels upon rescue. As the authors found more amino acids changing (please provide a statistics of whether the changes shown in Fig. S3A are significant) the authors should study of whether other pathways are affected through Phf8 decrease as well, and whether supplementation NSCs with reduced Phf8 expression with other amino acids, single or in combination, would restore cell proliferation to control levels.

22. If glycine feeds into purine synthesis, why ATP levels do not change, while CTP and TTP change as does GTP, which is the only change that makes the connection to purine synthesis plausible? How do the authors explain this finding? Maybe a closer look on the serine metabolism and other derivatives of serine (pyrimidines, folate) might be necessary to conceptualise this experiment correctly.

23. Are the replication errors as shown in Fig. 5E and F rescued upon supplementation with serine (or other amino acids)?

24. Fig. 6A: the blot does not show a clear band for LC3-II in non-treated samples - however it was set to an arbitrary value in the quantification and was even used for the statistical analysis. This is not convincing in its present form in the main figure.

25. Fig. 6C: two replicates are mentioned for this experiment. I wonder how the authors provide a solid statistics based on only two replicates? I am also sceptical about displaying only p-S6K and tubulin levels. Either the authors also show S6K levels in their quantifications in regard to tubulin, or they might consider to determine the total levels of S6K (pS6K+S6K), normalise this to tubulin and assess then the changes within the pS6K fraction.

26. "IPCs migrate to the subventricular zone (SVZ), downregulates the RGC marker SOX2 and upregulates the TBR2 (IPC marker),... grammar incorrect

27. The description of the IUE result might benefit from precision, as it is not clear, what the authors mean by "ectopic neurons". The quantifications provided do not allow unequivocally to exclude premature differentiation - to address this, BrdU labelling followed by determination of the leaving and the cycling fraction is needed. One can also envision a premature exit from the cell cycle followed by apoptosis because of a derailed metabolism and/or accumulating damaged DNA (based on the author's described experiments). In this context the difference of "aberrant neurogenesis" opposing a reduced "overall neurogenic output" should be rephrased to make clear what the authors want to address precisely - I at least do not understand which processes should collide here.

28. In the discussion the authors claim that "PHF8 triggers a transcriptional program characterized by the sequential repression of these genes.." - I think that the manuscript does not provide time-resolved transcriptional data that would allow claiming a sequential repression.

29. Further in the discussion, the authors state "...H3K9me2, our experiments did not show changes in H3K9me2/3 levels at serine biosynthesis gene promoters following PHF8 depletion." Indeed, the authors only show one immunostaining in regard to H3K9me2 and no data on H3K9me3. But not providing data does not justify an interpretation that H3K9me2/me3 levels do not change, specifically at promoters, which the authors did not study specifically by applying immunostainings.

ANSWER TO THE EDITOR AND THE REVIEWERS

EMBOR- EMBOR-2025-61935V1 "Epigenetic Regulation of Serine Biosynthesis by PHF8 During Neurogenesis"

We sincerely appreciate the constructive and insightful comments provided by the editor and reviewers. We have carefully addressed all the suggestions and made substantial efforts to improve the quality and robustness of our study. In particular, we have increased the number of biological replicates and added additional controls, performed new RNA-seq and ChIP-seq assays, and validated key findings by qPCR and Western blot analyses, including the use of a second shRNA. Moreover, we have strengthened our initial observations by incorporating new transcription factor analyses, additional *in vivo* experiments, and rescue assays. Altogether, these revisions further reinforce our conclusion that the regulation of serine by PHF8 contributes to the control of proliferation during neurogenesis.

We include below a detailed, "point-by-point" response to them. Reviewers' suggestions are printed in Times bold, and our replies are printed in Times. As a guide, however, we outline below the most important changes in the figures of the revised manuscript.

6 new figures (3 Figures EV and 3 Supplementary Figures) have been added.

MAIN FIGURES:

Figure 1: *Panel A* now includes a new analysis of PHF8 expression. *Panel D* presents new cell cycle data obtained from three independent replicates and new cell cycle markers.

Figure 2: *Panels A–E* have been modified and now include three new RNA-seq replicates. *Panel F* now contains a Western blot showing protein expression levels and a schematic overview of the experimental design. The previous *panel G* has been replaced with a new figure obtained from NSCs. The two graphs that were previously shown in *panel H* have been merged into a single graph.

Figure 3: *Panels C and E* have been updated to include the new RNA-seq data.

Figure 4: This corresponds to the former Figure 5, in which *panels B and C* have been combined into a single *panel (B)*. *Panel F* now presents a serine rescue experiment addressing DNA damage.

Figure 5: This corresponds to the former Figure 6. Panels A and C (Western blots) have been improved for image quality and clarity.

Figure 6: This corresponds to the former Figure 7, which remains unchanged.

FIGURES EV

Figure EV1: Contains data from the former *Appendix Figure S4* (Panels A–D). New data for PHF8 and H3K9me3 have been added.

Figure EV2: This corresponds to the former Figure 4. Panel B now includes a new analysis of TF expression during neurogenesis. Panel D adds a Western blot control for NFIA levels. Panel E has been updated to incorporate new RNA-seq data.

Figure EV3: A totally new figure confirming defects in proliferation and neurogenesis in PHF8-depleted mice, based on EdU labelling and Dcx immunofluorescence assays.

APPENDIX FIGURES

Appendix Figure S1: This figure presents new controls for shPHF8 RNA and additional data addressing the effect of PHF8 on cell cycle progression and proliferation.

Appendix Figure S2: This figure provides new control data derived from the updated RNA-seq analysis.

Appendix Figure S3: Includes *Panels C and D* from the former *Figure S2*. Adds new control panels (A and B) of the H4K20me1 ChIP-seq experiments.

Appendix Figure S4: A completely new figure showing no changes in H3K9me3 levels at SBP genes by ChIP-seq. The new *Panel F* corresponds to former Panel D from *Figure S2*.

Appendix Figure S5: A new figure presenting data on the contribution of ATF4 to SBF regulation and to NSC proliferation, and showing the proliferation profile upon NFIA and PHF8 depletion or overexpression.

Appendix Figure S6: Based on the former *Figure S3*. The heatmap (Panel A) has been replaced by a bar diagram including statistical information.

Appendix Figure S7: It is the former *Figure S4*. Now includes a new Panel B, showing a decrease in p-ULK levels in shPHF8 cells.

REVIEWER 1

We thank reviewer 1 for their constructive and positive comments. We acknowledge the significance of the raised issues and we agree about the possibility that other factors could cooperate with PHF8 or with PHF8 and NFIA to regulate SBP genes. In response to their recommendations, we have performed additional experiments to address them. Specifically, we have made an effort analyzing the potential role of ATF4.

1. The authors try to put NFIA and PHF8 in causal relation by showing a 26.3% overlap in altered gene expression. However, what does this mean? Can the authors benchmark this? How much % could one expect from a common pathway - or how much overlap would one expect when silencing two unrelated genes?

Response:

We thank the reviewer for this insightful comment. We agree that the initial 26.3% overlap in altered gene expression between NFIA and PHF8, while suggestive, was difficult to interpret without proper benchmarking. In the new RNA-seq analysis, the number of genes regulated by both proteins is 30%. Following the reviewer's suggestion, the revised figure now focuses only on genes that are downregulated in both PHF8 and NFIA knockdowns. Among these, 20.8% of the genes downregulated in PHF8 KD are also downregulated in NFIA KD (new Fig. EV2E).

To assess the likelihood of observing an overlap of 614 genes (20,8%) between the 2,946 downregulated in the PHF8 RNA-seq and the 2,408 genes downregulated in the NFIA RNA-seq by chance, we generated a random distribution of overlap values. This was done by comparing the 2,946 gene list with 10,000 random samples of 2,408 genes each. Specifically, we drew 10,000 random samples from the 21,810 mouse genes annotated in Ensembl (GRCm39), computed the overlaps, and constructed the resulting empirical distribution (Fig. EV2E). The observed overlap value (614) was not reached in any of the random samples, resulting in an empirical p-value of zero.

2. In light of the above mentioned, it would be interesting to compare with another known transcription factor (such as ATF4). Since mTORC1 activates the ATF4 axis, which in turn controls amino acid metabolism including serine biosynthesis and nucleotide metabolism

Response:

We fully agree with Reviewer 1; their comment is entirely justified. Following their recommendation, we analyzed the contribution of ATF4 in cooperation with PHF8. To this end, we used an shATF4 construct to reduce ATF4 levels and analyze the consequences in NSCs proliferation and to SBP gene expression. The results indicate

that, NFIA is the main contributor to the role of PHF8 in regulating NSC proliferation. The new results are now presented in Fig. EV2 and Appendix Fig. S5.

Minor comment:

In figure 1G, I would express all data in relation to shCtrl +Ser to show also the effect on SSP genes in shPHF8+ser vs shCtrl+Ser. On similar lines, I would merge the two panels in figure 1H as it would also show the lower proliferation rate in shPHF8 at baseline.

As suggested by the referee, both panels have been revised accordingly.

REFeree 2

We express our gratitude to Reviewer 2 for providing constructive and positive feedback, which I am confident has helped improve the quality of our work.

1. The Phf8 shRNA is a key tool in this study. However, almost no experiments were done to exclude the off-target effects of this shRNA. I suggest the authors repeat the NSC proliferation experiments (Figure 1C) with an shRNA targeting another sequence or test if the reduced proliferation could be rescued by overexpressing an shRNA-resistant Phf8 vector.

Response:

Referee is absolutely right. We used this shRNA because it has been extensively used and validated in our laboratory (Asensio-Juan *et al*, 2012; Iacobucci *et al*, 2021). Following the referee's recommendation, we repeated the proliferation assay using a second shRNA. The results obtained were similar. These new data are presented in Appendix Fig. S1C.

2. Does Phf8 gain-of-function increase NSC proliferation?

Response:

Following referee's suggestion, we have overexpressed PHF8 in NSCs and the proliferation status have been evaluated. No changes in proliferation have been identified. This new result is displayed in Appendix Figure S1E.

3. No evidence for knockdown of protein was shown for Nf1a shRNA. What is the relationship between Phf8 and Nf1a? Would double knockdown additionally reduce NSC proliferation? Would the reduced NSC proliferation induced by knockdown of Phf8 or Nf1a be reversed by overexpression of the other one?

4. As related to 3, why were Nrf1, Afx, Atf4, E2f4, etc. not investigated for their roles in coordinating with Phf8 to regulate SBP gene expression and NSC proliferation considering their motif enrichment is higher

than Nf1a? The authors' statement that they have previously demonstrated Phf8 regulates Nf1a expression does not justify this scientifically.

Response to 3 and 4:

We focused on *Nfia* because, in a gene expression analysis of the mouse brain (Yao *et al.*, 2023), NFIA showed the highest expression levels among the stages analyzed—those in which PHF8 is also most strongly expressed (new Fig. EV2B). Moreover, it has been recently reported that NFIA regulates chromatin opening in a temporally dependent manner during vertebrate neural tube development (Zhang *et al.*, 2025), thereby influencing posterior differentiation. Furthermore, NFIA has been reported to be directly regulated by PHF8 in several biological contexts (Iacobucci *et al.*, 2021). This observation suggested the possibility of a regulatory feedback loop that could contribute to PHF8-mediated sensing of intracellular amino acid levels, an idea that seemed appealing to us. However, we agree with the reviewer that this alone is not sufficient to demonstrate a functional collaboration between the two factors. As the reviewer correctly notes, other transcription factors may also participate in this regulatory network. In particular, ATF4 stands out due to its known activation by mTORC1, its well-established role in amino acid metabolism, and its expression at the developmental stages analyzed.

Following the reviewer's suggestions, we first confirmed the NFIA KD by Immunoblot (Fig. EV2D). Next, we further investigated the relationship between NFIA and PHF8 by performing additional experiments, including the depletion of both factors and the assessment of their effects on cell proliferation, as well as PHF8 overexpression in NFIA-knockdown cells (Appendix Fig. S5A). We also explored the potential contribution of ATF4 in coordinating with PHF8 response (Appendix Fig. S5B-E). The new results indicate that NFIA is the main contributor to the role of PHF8 in regulating NSC proliferation.

5. The authors stated that 65% of genes that changed the accessibility upon Phf8 depletion also suffered changes in transcription in the RNA-seq (Figure 3E). First, how was 65% calculated? Second, are changes in chromatin accessibility and gene expression in the same direction? If not, this could be easily interpreted in the opposite way. The second question also applies to Figure 4D.

Response:

We apologize for the error. The referee is correct — the accurate value is not 65%. This has been corrected and updated based on results from new replicates of the PHF8 RNA-seq analysis. Furthermore, the figure now displays only downregulated genes with decreased chromatin accessibility.

Regarding Figure 4D (now Fig. EV2E), it has been modified to include only downregulated genes in both the PHF8 and NFIA RNA-seq datasets. The corresponding text has been revised accordingly to clarify this point.

6. Several previously published sequencing datasets used for analysis in this study were not obtained from NSCs, but were used by the authors to directly compare with their NSC data, which raises the concern of tissue specificity or cell type specificity. This should at least be discussed.

Response:

The referee is absolutely right. We have made an effort to use sequencing datasets derived from the same cell type as ours; however, in a few cases (Fig. 3A, Fig. EV2E and Appendix Fig. S5D) where no published data were available for NSCs, we resorted to using data from functionally and developmentally related cell types. We fully acknowledge that these datasets are not strictly comparable, as tissue-specific effects may influence the results. Therefore, the findings should not be interpreted literally, but rather as indicative of a potential involvement of PHF8 in the analyzed function. In line with the referee's suggestion, this limitation has now been explicitly addressed in the revised manuscript.

7. In Figure 6A, the representative blot does not seem to represent the quantification result well.

Response:

Following referee indication, the blots have been changes to better represent the results

Minor:

1. The source of the single-cell RNA-seq that was used for analysis in Figure 1A was not stated in the text or the appendix table.

Response:

The link to the website, as well as the reference, have been included in the main text (page 5).

2. Please explain in the manuscript how Phf2 and Kiaa1718 are related to Phf8.

Response:

PHF8, PHF2 and KIAA1718 are members of the KDM7 subfamily. The three enzymes are capable of demethylating mono- and dimethylated lysine residues, but not trimethylated ones. Structurally, they are highly similar, as they contain both a JmjC domain and a PHD finger domain, which mediates recognition of H3K4me3 (Chaturvedi *et al*, 2019; Fueyo *et al*, 2015). Functional interplay between PHF2 and PHF8 has been reported (Shi *et al*, 2014). In response to the referee's suggestion, the relationship between these two proteins has now been clarified and explicitly described in the revised manuscript.

3. Figure S1F, please clearly state what data were used for analysis.

Response:

We thank the reviewer for pointing out the omission. We apologize for not having previously indicated the source of the data presented in Figure S1F. As clarified in the revised manuscript (page 8), these data were obtained from our own RNA-seq experiments.

4. Figure 2G, please make sure the comparisons are correct.

Response:

We appreciate the reviewer's observation. We acknowledge that the original comparisons were not appropriate. The experiment has been repeated in NSCs according to the reviewer's suggestions, and the updated figure now correctly represents the intended comparisons.

5. Figure 2H, does NEAA mean no NEAA?

Response:

We thank the referee for the comment, which allows us to clarify the annotation: "-NEAA" means the absence of non-essential amino acids (NEAAs have been removed from the medium). For clarity, we have replaced the "-" symbol in the figure with "w/o" in the revised version.

6. Why was $p < 0.08$ instead of 0.05 used as the cut off in RNA-seq analysis?

Response:

We appreciate the reviewer's comment and agree that the rationale for using a p-value cutoff of 0.08 was not clearly explained. Initially, we performed the RNA-seq analysis using both the conventional cutoff of 0.05 and a more permissive threshold of 0.08. We chose to further analyze the 0.08 dataset because several genes validated by qPCR—many of which are involved in metabolic processes—were not captured at the stricter 0.05 threshold but were included at 0.08.

However, the RNA-seq experiments and subsequent analyses have now been repeated, and in the revised version we consistently apply $p < 0.05$ as the cutoff (see Fig. 2).

7. The authors declared there is a global increase in bulk methylation following Phf8 knockdown, however this does not seem to be the case for H3K9me2.

Response:

The referee is absolutely right. Our original conclusion that global methylation increases is not accurate for all histone marks. While several marks we analyzed do show an increase (some of which were not included in the first version of the manuscript), this is not true for all, including H3K9me2. Consistent with our previous observations in other neural contexts (Iacobucci *et al.*, 2021), we found that H3K9me3 levels increase globally after PHF8 depletion. We interpret that the lack of increase in H3K9me2 is because it serves as a substrate for SUV39H1/2 enzymes, which convert H3K9me2 into H3K9me3, the final product. We have now included

immunostaining for H3K9me3 (Fig. EV1D) in the revised manuscript and rewritten the related text to better reflect these results.

8. Why was a decrease on H4K20me1 levels at the SBP gene bodies upon Phf8 depletion expected due to downregulation of these genes? Please explain and include citations as needed.

Response:

We thank the referee for this comment, which allows us to better explain the possible contribution of a relatively understudied epigenetic mark, H4K20me1. It has been proposed that H4K20me1 at promoters plays a repressive role (Asensio-Juan *et al.*, 2012; Beck *et al.*, 2012); however, its presence along the gene body is associated with transcriptional activation and elongation (Shoaib *et al.*, 2021). Consistent with this, in our presented results, following PHF8 depletion, a slight increase of H4K20me1 is observed at promoters, while a depletion occurs along gene bodies. New references supporting this observation have been included in the revised version of the manuscript.

9. Methods-Plasmids and recombinant proteins, in what experiment(s) was the pInducer vector expressing wild type Phf8 used?

Response:

Thank you for pointing out this unclear aspect in our original version. The pInducer expressing PHF8 WT was used in Fig. 2F to rescue the transcriptional defects observed following PHF8 depletion, and in Appendix Fig. S1E and Appendix Fig. S5A to overexpress PHF8. This has now been clarified in the revised version of the manuscript.

10. Methods, should the transposase be Tn5 instead of Th5?

Response:

We apologize for the error pointed out by the referee. It has been corrected in the revised version of the manuscript.

REFEREE 3

I would like to express my gratitude to Referee 3 for their constructive comments, which have prompted us to reflect on certain aspects of our work, particularly the made more robust our data and modulate the interpretation of the results. In the revised version of the manuscript, we have made a significant effort to strengthen the robustness of our data by increasing the number of replicates, (particularly in the RNA-seq experiments), adding additional controls, and validated key findings by qPCR, and Western blot analyses. Moreover, we have strengthened our initial observations by incorporating new transcription factor (ATF-4) analyses, additional *in*

vivo experiments, and rescue assays to reinforce the central role of PHF8 in regulating proliferation during neurogenesis through metabolic control.

We have also modified the discussion incorporating a broader perspective about the role of PHF8 regulating proliferation during neurogenesis to reflect the complexity and potential interplay of mechanisms underlying PHF8-dependent phenotypes.

I am confident that these comments have contributed to the overall improvement of our work, and we wish to extend our thanks.

GENERAL COMMENTS

There might be different layers towards the phenotypic observations, and the authors might take this into account by judging of whether an entire novel mechanism is provided

Response:

We would like to thank the referee for this insightful comment, with which we fully agree. In this study, we focused on the metabolic consequences of partial PHF8 loss and their impact on proliferation and neurogenesis. However, we acknowledge that other factors may also contribute to the impaired proliferation and altered neurogenesis observed. Cell cycle-related genes consistently appear across our analyses, and their role in cell cycle progression (Lim *et al*, 2013; Liu *et al*, 2010; Sun *et al*, 2015), cytoskeletal dynamics, (Asensio-Juan *et al.*, 2012) and DNA repair (Cheng *et al*, 2020; Kim *et al*, 2024; Lee *et al*, 2015) has been described both by us and others in various cellular models. These alterations, independent of or parallel to metabolic changes, may influence the final phenotypic outcome. With our study, we aim to highlight that, in addition to previously described regulatory mechanisms, we now report for the first time, metabolic alterations mediated by PHF8 that contribute to the proliferation defects. Furthermore, this is also the first description of this phenotype *in vivo*, in mammals.

Following the referee's suggestion, we have now incorporated this broader perspective into the discussion (pages 21, 22) to reflect the complexity and potential interplay of mechanisms underlying PHF8-dependent phenotypes.

Following up on this, some of the experiments do not have the number of replicates necessary to provide statistical analysis, especially the RNAseq data, but also other data are shown with numbers of replicates (two biological replicates) that do not allow a valid statistical testing

Response:

We agree with Reviewer 2's comment. While most of the experiments were originally performed using three biological replicates, some results were based on only two. In the revised version of the manuscript, we have made a significant effort to strengthen the robustness of our data by increasing the number of replicates,

particularly in the RNA-seq experiments, Western blot analyses, and qPCR assays. We further confirmed the RNA-seq and ChIP-seq results through qPCR validations. Additionally, we have confirmed key findings using an independent shRNA construct.

COMMENTS

1. "...processes that occur in parallel with extensive epigenomic remodeling driven by histone acetyltransferases (KATs), histone methyltransferases (KMTs), and histone demethylases (KDMs). Among the former, PHF8, ..." - it is probably more the latter than the former

Response:

The referee is right. The mistake has been corrected.

2. Fig. 1A: Expression over developmental time points is interesting, but should be enriched by untangling the specific cell types, and ordered in pseudotime to display the developmental dynamics in Phf8 expression in cellular distribution. From such an analysis a rationale to focus on NSCs can be derived by the authors, as a rationale is as of yet not given.

Response:

Following the reviewer's recommendation, we have revised the corresponding figure and expanded our analysis to better distinguish PHF8 expression across specific cell types over developmental time points. Our new data, reveal that PHF8 is predominantly expressed in apical and intermediate progenitor cells, during stages E12 to E16 in the ventricular zone. This finding provides a rationale for focusing our functional studies on NSCs using a mouse embryonic model at stage E12.5, which is mainly composed of neural progenitor cells that undergo intense proliferation (Dennis *et al*, 2016; Tiberi *et al*, 2012).

If Phf8 is also expressed in neurons, it might be interesting to study exemplarily its role in neurons, as these cells switch their metabolism substantially compared to NSCs. If Phf8 specifically impacts amino acid metabolism in NSCs, but not in neurons, these cell-specific functions in regulation metabolism will spice up the findings of the authors.

Response:

We greatly appreciate the reviewer's valuable comment that could bring more interest to our findings. However, investigating the role of PHF8 in neurons is beyond the scope of the present study. Addressing this at the

molecular level would require optimizing an in vitro neuronal differentiation system, which is not easy to established. Furthermore, as the reviewer pointed out, metabolism differs significantly between cell lineages, then, even if we observe metabolic differences, the role of these metabolic changes could still differ due to the distinct physiology and metabolic demands of the two cell types. For example, serine is involved in neurotransmitter synthesis in neurons (El-Hattab, 2016), or glycine is working as a neurotransmitter itself (Bowery & Smart, 2006; Lopez-Corcuera *et al*, 2001) highlighting cell type-specific metabolic roles that may not be reflected solely by metabolite' levels.

3. The KD efficiency of Phf8 is around 50%. The authors might want to bring this in the context of whether this model has relevance to the human disease. As shRNAs can have variable and off-side effects, I recommend to use a genetic deletion as additional model to substantiate the findings. In their earlier publication (Iacobucci et al 2021) a similar approach has been used to study Phf8 in astrocytes; to advance towards a gain of specificity using for example a CRISPR/Cas9 approach to generate a genetic model might be of advantage.

Response:

We appreciate the reviewer's comment. In fact, we are currently exploring an alternative model to de shRNA in the lab; however, this has proven technically challenging. Generating CRISPR-mediated knockouts in NSCs is particularly difficult due to the limitations in performing clonal selection.

Additionally, we would like to mention that achieving depletion levels above 50–60% often leads to cell death or senescence, which makes downstream experiments unfeasible. We also attempted to develop an inducible system using the Tet-On/Off system under doxycycline control. However, this approach was not suitable, as the induction process had strong adverse effects on NSCs metabolism.

Although the knockdown system used in this study has been extensively validated and widely applied in our laboratory (Asensio-Juan *et al*, 2017; Asensio-Juan *et al.*, 2012; Iacobucci *et al.*, 2021), to confirm the specificity of the observed proliferation effects, we performed a rescue experiment by overexpressing PHF8 (Fig. 2F). Moreover, following Reviewers' recommendation, we show that similar effects on proliferation are observed using a second independent shRNA (Appendix Fig. S1C).

4. The cell cycle analysis requires more substance in order to be interpretable - as of yet, the interpretations of the authors are based on non-significant statistical results. I suggest to increase the number of replicates and to provide experimental data on differences in cells leaving or re-initiating the cell cycle. What is the

reason for a potential delay in S-phase entry, if this is indeed the main difference upon Phf8 KD? Taking the later finding into account that the replication fork stalls upon Phf8 KD - would this not rather result in S-phase progression defects? Here, clarification is needed benefitting the coherence of the data presented.

Response:

We thank the reviewer for this insightful comment, which helped us clarify our interpretation of the role of PHF8 in cell cycle progression. As suggested, we performed additional replicates of the cell cycle analysis. Moreover, we now include a comprehensive evaluation using PI staining, BrdU incorporation (to assess active S phase), and MPM2 (a mitotic marker). These new results are presented in Figure 1D and Appendix Figure S1D. PI analysis indicates that Phf8 knockdown (KD) does not lead to cell cycle exit; however, it results in a significant decrease in the proportion of S-phase cells, accompanied by a slight increase in G1-phase cells. Importantly, the reduction in S-phase cells detected by PI is corroborated by a significant decrease in BrdU-positive cells, confirming impaired DNA synthesis. In addition, the significant reduction in MPM2-positive cells upon Phf8 KD supports the notion that these cells may accumulate DNA damage during S phase and consequently require additional time to repair this damage during G2 before entering mitosis.

Taken together, these new results clearly demonstrate that Phf8 KD cells exhibit defects in cell cycle progression. In light of the DNA fiber assay data presented later in the manuscript, we conclude that these defects arise in part, from deficiencies in replication fork progression.

5. It is not sufficient to list the packages / tools used for bioinformatics analysis, but it is necessary to provide the versions and the analysis settings in order to adhere to basic principles allowing to reproduce the data. I am missing an accessible repository with all bioinformatics workflows that will allow reproducing the analyses.

Response:

In response to the referee's suggestion, we have now included additional details regarding the bioinformatics tools used in our analyses. All tools are publicly available and have been widely validated. The specific versions used are now clearly indicated in the Methods section.

6. Along the line of comment #4, it is surprising that DESeq2 tools can handle 1 ctrl and 2 shRNA KD samples to return data of significance, and that the authors can provide a p-value. The RNA-seq experiment is unacceptable, as it is not based on an appropriate numbers of biological replicates for both conditions that

are compared. Following ENCODE state-of-the-art, the authors need to improve this experiment and bring it to current standards.

Response:

The referee is absolutely right, and we apologize for the initial inclusion of only a single control sample in the RNA-seq experiment. Following their recommendation—as well as the ENCODE guidelines—we have repeated the RNA-seq experiment using three biological replicates for both control and PHF8 knockdown cells (Figure 2). Additionally, we have validated the results by qPCR (Appendix Figure S2B) and have provided a new summary table compiling the results from both the RNA-seq and ChIP-seq analyses (Appendix Table S2), which illustrates the consistency of our data. The results obtained are consistent with those presented in the previous version, particularly regarding the regulation of metabolic genes, including those involved in serine metabolism. These new data further support our conclusion that PHF8 functions as a transcriptional regulator of genes involved in serine metabolism.

7. The experimental setup of the rescue experiment in Fig. 2F is not given. The degree of OE in the conditions of presence/absence of shRNA should be assessed by immunoblots of PHF8, to provide the levels of PHF8 present in the cell. Seemingly, the OE results in a higher transcriptional response compared to control levels. In addition, the authors should provide a scheme in the supplement illustrating the rescue strategy, where the shRNA is targeting and whether this sequence is missing in the OE plasmid.

Response:

Following the referee's suggestion, we have performed immunoblot to assess PHF8 protein levels upon overexpression (Figure 2F). In addition, we have included a schematic representation in the Figure 2F (top panel) to illustrate the rescue strategy. It is important to note that the OE plasmid contains the sequence targeted by the shRNA. However, overexpression of wild-type PHF8 can compensate for or override the effects of the shRNA, as evidenced by the increased PHF8 mRNA levels (qPCR) and protein levels (immunoblot) observed in the rescue condition.

8. Fig. 2G might be moved to a supplement to illustrate that HeLa cells respond to serine-depletion. However, the authors should indicate precisely what has been compared to what in their statistics - each gene has to be compared to its counterpart; the way of representation in its present form is misleading - it might suggest that all conditions might be statistical significant, which I doubt is the case here. As the authors used the Student's t-test, some more parameters should be given (one- or two-sided?) and the authors should test the normal distribution of their data.

9. A new experiment analogous to Fig. 2G using NSCs should be provided. Here, the levels of the control experiment should be compared to the shPhf8 condition as well. And the serine condition should be tested as further variable to discriminate between shPhf8 effects and serine effects.

Response to 8 and 9 comments:

We thank the referee for their thoughtful comments. We would like to clarify that the primary aim of this figure is to assess whether the transcriptional response to serine deprivation is attenuated in PHF8 knockdown cells compared to control cells. For this reason, each line was normalized to its respective control.

Nonetheless, in line with the referee's recommendation, we have repeated the analysis in NSCs to include a direct comparison between control and shPHF8 conditions. The data were confirmed to follow a normal distribution (Shapiro-Wilk test), and two-tailed Student's t-tests were applied accordingly. The results show, consistent with our observations in HeLa cells, that shPHF8 cells fail to activate SBP genes in the absence of serine, in contrast to shCTR cells. These clarifications have been introduced in the revised version of the manuscript. To ensure consistency across the manuscript, Figure 2G has been replaced with data from NSCs, and the HeLa cell data have been removed.

10. Fig. 2H should be combined to illustrate both conditions together and to give indication of the robustness of shPhf8 KD effect compared to control. The accompanying Fig. S1E shows a milder effect, and not as the authors claim a similar effect, in terms of statistical significance and number of cells. Does the proliferation cease completely if a combination treatment of shPhf8 and PHGDH inhibition is applied?

Response:

We thank the reviewer for this comment. The reason for presenting two separate graphs was to more clearly show that the addition of NEAA has no effect on control cells, but does impact PHF8 KD cells. As indicated by the graph scales and described in the text, the rescue is not complete—i.e., amino acid supplementation does not fully restore proliferation to control levels—but it does lead to a noticeable improvement in PHF8 KD cells, unlike in control cells. This result is consistent with our expectations. In addition to amino acid availability, other metabolic factors and gene expression changes in PHF8-depleted cells likely contribute to the observed effects on cell proliferation. These data are intended to emphasize that serine alterations play a contributing role, though they are not the sole drivers, of the proliferation defects following PHF8 depletion.

Following the reviewer's recommendation, we have modified the figure.

The sentence related to the former Fig. S1E (new Appendix Figure S2D) has been modified to better reflect the results.

11. Fig. S2A needs to include expression of PHF8.

Response:

Following the referee's suggestion, we have included a new panel showing the expression levels of PHF8 in control and PHF8 KD cells (Fig. EV1A).

And levels of the H4K20me1 and H3K9me2 need to be shown in immunoblots together with the respective quantification.

Following the referee's suggestion, we quantified histone methylation changes by immunoblot (see Figure Referee 3). However, the results obtained for H4K20me1 were inconsistent and varied between experiments; therefore, we decided not to include them. We would like to emphasize that the immunofluorescence images were analyzed as quantitatively as possible, and the observed differences—particularly for H4K20me1 and H3K9me3—were sufficiently evident to support our conclusions. To ensure the most quantitative comparison of the immunofluorescence images, they were acquired with identical acquisition settings (laser power, detector gain, exposure time) for all experimental conditions within each replicate. Secondary antibody-only samples were included as negative controls to assess non-specific signal. For quantification, 15 random fields per biological replicate ($n = 3$ independent experiments) were imaged, and mean fluorescence intensity per cell was measured using ImageJ. More than 100 cells per condition were analyzed. Nuclei were segmented based on DAPI staining, and cell masks were generated to define measurement regions. Background fluorescence was subtracted using secondary antibody-only controls, and intensity values were normalized to cell area. The threshold for positive signal was defined as the mean + 2 standard deviations of the negative control.

12. Fig. 3B: the authors should provide a k-means clustering with a heatmap illustrating peak enrichment genome-wide, in ctrl and KD conditions.

Response:

Following the referee's suggestion, we have included the heat maps as well as additional ChIP-seq controls, including validation by qPCR, (Appendix Figure S3A, B), and the Appendix Table S2 showing the reads obtained and analyzed in these experiments.

13. Describing the results of Fig. 3B the authors refer to GO analysis of the regions - GO terms rely on genes that are usually referred to as the ones lying nearest to the peak region. Please describe more exactly.

Response:

We thank the reviewer for this insightful comment, which prompted us to clarify the details of our analysis. The GO analysis shown previously in Appendix Fig. S2B (now Appendix Fig. S3C) was performed using genes located closest to H4K20me1 peak regions that were higher in KD cells compared to control cells. This clarification has now been incorporated into the revised version.

14. "Among these, the SBP genes (Fig. 3C and Appendix Fig.S2C)." This is not a full sentence.

Response:

The error has been corrected, and the sentence has been replaced with: 'Among these, some SBP genes were identified'

15. The authors claim that H4K20me1 is the main target of Phf8 in NSCs, based on immunostainings as provided in Fig. S2A. As mentioned above, this finding should be backed up with immunoblots as quantitative method. However, the authors claim to see increase of H4K20me1 in promoter regions, alongside decrease in gene bodies. Thus, the global view point as provided for H3K9me2 might hide changes that go in opposite directions at different locations in the genome as well. Therefore, the interpretation of a main and a minor target might be premature, unless the authors provide clear experimental evidence. H3K9me2 ChIP-seq would be an appropriate means.

Response:

We thank the referee for this insightful comment. Following the suggestion, we analyzed H3K9me2 levels by immunoblot (see Figure Referee 3) and performed three independent H3K9me2 ChIP-seq experiments. However, the enrichments observed in our positive controls indicated that these ChIP-seq experiments did not perform reliably, and we have not yet determined the cause of this issue.

Given that H3K9me2 is a substrate of Suv39h1/2, which are capable of using it to generate H3K9me3, we hypothesized that the H3K9me2 produced could be further methylated, leading to increased levels of H3K9me3. To test this, we analyzed H3K9me3 levels both by immunofluorescence (Figure EV1D) and immunoblot (See Figure Referee 3). Our results showed a global increase in H3K9me3. Based on these findings, we performed ChIP-seq for H3K9me3, and the results are included in the new Appendix Figure S4. Despite the global increase, no changes were observed in H3K9me3 methylation levels at the promoters of genes involved in the serine biosynthesis pathway. These results further support the role of H4K20me1 as a PHF8-mediated mark involved in the regulation of serine biosynthesis genes. It is worth highlighting that, in our hands, the neural context—particularly in NSCs—H4K20me1 appears to be the predominant regulatory mark (Asensio-Juan *et al.*, 2012; Iacobucci *et al.*, 2021). Although we currently do not have an explanation for this specificity, it underscores the potential importance of H4K20me1 in neural epigenetic regulation.

Following the referee's recommendation, we have moderated our conclusions regarding the exclusive role of H4K20me1 in regulating these genes.

16. " This assays assesses" grammar incorrect

Response:

We apologize for the mistake. The error has been corrected

17. Fig. 3E: the authors state that 65% of genes with differentially accessible regions are found among differentially expressed genes. However, the numbers given in the figure do not match 65%. The authors should clarify this and provide the numbers underlying the 65% they claim to see, while describing the results in the text.

Response:

The referee is absolutely right, and we sincerely apologize for the error in the calculation. The figure has been corrected and updated based on results from new replicates of the PHF8 RNA-seq analysis. In addition, it now includes only downregulated genes with decreased chromatin accessibility, as suggested by Referee 2.

18. " Thus, we evaluate the impact of NFIA depletion" inconsistent grammar

Response:

The sentence has been substituted with: "Thus, we evaluated the impact of NFIA depletion"

19. " GO of these genes revealed multiple categories" unprecise phrasing

Response:

We apologize for the imprecise wording, which has now been revised to: "GO analysis of these genes identified functional categories related to cell cycle..."

20. "similar or even higher to the observed upon PHF8 depletion (Fig. 4F)." incomplete sentence

Response:

The error has been corrected, and the sentence has been completed as follows: "similar to or even higher than that observed upon PHF8 depletion".

21. Fig. 5B: both conditions of shRNA treatments should be represented in one figure to assess whether serine supplementation would reach ctrl levels upon rescue. As the authors found more amino acids changing (please provide a statistics of whether the changes shown in Fig. S3A are significant) the authors should study of whether other pathways are affected through Phf8 decrease as well, and whether supplementation NSCs with reduced Phf8 expression with other amino acids, single or in combination, would restore cell proliferation to control levels.

Response:

We thank the reviewer for this comment. Please also see our response to comment 10.

The reason for presenting two separate graphs was to better illustrate that serine supplementation has distinct effects on control versus PHF8 KD cells. As shown in the graphs, serine supplementation does not fully rescue proliferation in PHF8 KD cells. This result is expected, as PHF8-depleted cells also exhibit alterations in other amino acids and metabolites besides serine and changes in gene expression that affect proliferation (see response to comment 10 and 4). Our intention with these data was to highlight that the increase in serine levels supports growth in PHF8 KD cells, unlike in control cells. Following the reviewer's suggestion, the figure has been modified accordingly, and this point is now addressed in the revised Discussion.

All differences shown in the updated Appendix Figure S6A are statistically significant.

It is possible that other metabolic pathways contribute to the observed effects. However, we focused on serine because, based on our RNA-seq data, it was the only amino acid whose biosynthetic enzymes appeared to be directly regulated by PHF8. The study of additional pathways is beyond the scope of the present work, but we consider this an interesting direction for future research.

22. If glycine feeds into purine synthesis, why ATP levels do not change, while CTP and TTP change as does GTP, which is the only change that makes the connection to purine synthesis plausible? How do the authors explain this finding? Maybe a closer look on the serine metabolism and other derivatives of serine (pyrimidines, folate) might be necessary to conceptualise this experiment correctly.

Response:

We appreciate the reviewer's comment regarding the role of serine in nucleotide biosynthesis and apologize for any lack of clarity in the previous version. While serine is not directly incorporated into nitrogenous bases, it supports nucleotide synthesis in two key ways. First, it serves as a precursor to glycine, which is directly incorporated into purine formation. Additionally, when converted to glycine, serine donates one-carbon units via N⁵,N¹⁰-methylenetetrahydrofolate (CH₂-THF), which are essential for both purine ring construction and the conversion of dUMP to dTMP in pyrimidine synthesis. Thus, serine supports cellular nucleotide pools in an indirect yet indispensable manner, particularly in rapidly proliferating cells. We have clarified these points in the revised manuscript (page 14 and Fig. 4C).

23. Are the replication errors as shown in Fig. 5E and F rescued upon supplementation with serine (or other amino acids)?

Response:

Following the referee's recommendation, we analyzed the effect of serine or amino acids supplementation on DNA damage accumulation using immunostaining assays. The results show that serine addition rescued the DNA damage. These data are presented in the new Figure 4F.

24. Fig. 6A: the blot does not show a clear band for LC3-II in non-treated samples - however it was set to an arbitrary value in the quantification and was even used for the statistical analysis. This is not convincing in its present form in the main figure.

Response:

We apologize to the referee for the lack of clarity in the original figure. The Western blots have been repeated, and a representative image that better reflects the results has now been included in the revised Figure 5A.

25. Fig. 6C: two replicates are mentioned for this experiment. I wonder how the authors provide a solid statistics based on only two replicates? I am also sceptical about displaying only p-S6K and tubulin levels.

Either the authors also show S6K levels in their quantifications in regard to tubulin, or they might consider to determine the total levels of S6K (pS6K+S6K), normalise this to tubulin and assess then the changes within the pS6K fraction.

Response:

Following the referee's recommendations, we included additional replicates of the Western blot to ensure robust statistical analysis. The quantification graph has been normalized to total S6K, as suggested (see Fig. 5C).

26. "IPCs migrate to the subventricular zone (SVZ), downregulates the RGC marker SOX2 and upregulates the TBR2 (IPC marker),... grammar incorrect

Response:

We apologize for the error, which has been corrected as follows: "IPs migrate to the subventricular zone (SVZ), downregulate the RGC marker SOX2, and upregulate TBR2, an IP marker"

27. The description of the IUE result might benefit from precision, as it is not clear, what the authors mean by "ectopic neurons". The quantifications provided do not allow unequivocally to exclude premature differentiation - to address this, BrdU labelling followed by determination of the leaving and the cycling fraction is needed. One can also envision a premature exit from the cell cycle followed by apoptosis because of a derailed metabolism and/or accumulating damaged DNA (based on the author's described experiments). In this context the difference of "aberrant neurogenesis" opposing a reduced "overall neurogenic output" should be rephrased to make clear what the authors want to address precisely - I at least do not understand which processes should collide here.

Response:

We appreciate the reviewer's insightful comments. We have revised the description of the IUE results for greater precision. To address the concern regarding premature differentiation of RGP, we performed EdU labeling of electroporated progenitors. The results, now shown in the new Figure EV3, reveal a higher rate of EdU retention and a reduced cell cycle leaving fraction in shPHF8 cells compared with controls, consistent with slower rates of cell cycle exit. Furthermore, the observed reduction in TUJ-positive cells supports the conclusion that shPHF8 progenitors exhibit delayed neurogenic output due to impaired progression through the cell cycle. In addition, we analyzed the expression of the early neuronal marker DCX, which is normally expressed in migratory neurons. Notably, DCX expression was absent in cells derived from shPHF8 progenitors, indicating defective neuronal differentiation rather than aberrant or ectopic neurogenesis. However, based on our data, we cannot rule out apoptosis of neurons that fail to differentiate in vivo.

Together, these data clarify that the phenotype reflects a reduction in neurogenic output due to impaired progenitor maturation and differentiation, rather than premature differentiation.

28. In the discussion the authors claim that "PHF8 triggers a transcriptional program characterized by the sequential repression of these genes.." - I think that the manuscript does not provide time-resolved transcriptional data that would allow claiming a sequential repression.

Response:

The referee is absolutely right — a time-course analysis was not performed. Consequently, we have removed the word 'sequential' from the sentence

29. Further in the discussion, the authors state "...H3K9me2, our experiments did not show changes in H3K9me2/3 levels at serine biosynthesis gene promoters following PHF8 depletion." Indeed, the authors only show one immunostaining in regard to H3K9me2 and no data on H3K9me3. But not providing data does not justify an interpretation that H3K9me2/me3 levels do not change, specifically at promoters, which the authors did not study specifically by applying immunostainings.

Response:

The referee is completely right. We agree that the original statement was overstated, as the first version of our manuscript did not include data specifically examining H3K9me2 or H3K9me3 at the promoters of serine biosynthesis genes. In the revised manuscript, we have now included H3K9me3 data, which indicate that PHF8-mediated H3K9me3 modification does not play a key role in regulating these genes (see also our response to comment 15). Accordingly, the sentence in the Discussion has been revised to reflect these new results.

REFERENCES

Asensio-Juan E, Fueyo R, Pappa S, Iacobucci S, Badosa C, Lois S, Balada M, Bosch-Presegue L, Vaquero A, Gutierrez S *et al* (2017) The histone demethylase PHF8 is a molecular safeguard of the IFN γ response. *Nucleic Acids Res* 45: 3800-3811

Asensio-Juan E, Gallego C, Martinez-Balbas MA (2012) The histone demethylase PHF8 is essential for cytoskeleton dynamics. *Nucleic Acids Res* 40: 9429-9440

Beck DB, Oda H, Shen SS, Reinberg D (2012) PR-Set7 and H4K20me1: at the crossroads of genome integrity, cell cycle, chromosome condensation, and transcription. *Genes Dev* 26: 325-337

Bowery NG, Smart TG (2006) GABA and glycine as neurotransmitters: a brief history. *Br J Pharmacol* 147 Suppl 1: S109-119

Chaturvedi SS, Ramanan R, Waheed SO, Karabencheva-Christova TG, Christov CZ (2019) Structure-function relationships in KDM7 histone demethylases. *Adv Protein Chem Struct Biol* 117: 113-125

Cheng Y, Liu N, Yang C, Jiang J, Zhao J, Zhao G, Chen F, Zhao H, Li Y (2020) MicroRNA-383 inhibits proliferation, migration, and invasion in hepatocellular carcinoma cells by targeting PHF8. *Mol Genet Genomic Med* 8: e1272

Dennis D, Picketts D, Slack RS, Schuurmans C (2016) Forebrain neurogenesis: From embryo to adult. *Trends Dev Biol* 9: 77-90

El-Hattab AW (2016) Serine biosynthesis and transport defects. *Mol Genet Metab* 118: 153-159

Fueyo R, Garcia MA, Martinez-Balbas MA (2015) Jumonji family histone demethylases in neural development. *Cell Tissue Res* 359: 87-98

Iacobucci S, Padilla N, Gabrielli M, Navarro C, Lombardi M, Vicioso-Mantis M, Verderio C, de la Cruz X, Martinez-Balbas MA (2021) The histone demethylase PHF8 regulates astrocyte differentiation and function. *Development* 148

Kim JE, Pan X, Tse KY, Chan HH, Dong C, Huen MSY (2024) PHF8 facilitates transcription recovery following DNA double-strand break repair. *Nucleic Acids Res* 52: 10297-10310

Lee C, Hong S, Lee MH, Koo HS (2015) A PHF8 homolog in *C. elegans* promotes DNA repair via homologous recombination. *PLoS One* 10: e0123865

Lim HJ, Dimova NV, Tan MK, Sigoillot FD, King RW, Shi Y (2013) The G2/M Regulator Histone Demethylase PHF8 Is Targeted for Degradation by the Anaphase-Promoting Complex Containing CDC20. *Mol Cell Biol* 33: 4166-4180

Liu W, Tanasa B, Tyurina OV, Zhou TY, Gassmann R, Liu WT, Ohgi KA, Benner C, Garcia-Bassets I, Aggarwal AK *et al* (2010) PHF8 mediates histone H4 lysine 20 demethylation events involved in cell cycle progression. *Nature* 466: 508-512

Lopez-Corcuera B, Geerlings A, Aragon C (2001) Glycine neurotransmitter transporters: an update. *Mol Membr Biol* 18: 13-20

Shi G, Wu M, Fang L, Yu F, Cheng S, Li J, Du JX, Wong J (2014) PHD finger protein 2 (PHF2) represses ribosomal RNA gene transcription by antagonizing PHF finger protein 8 (PHF8) and recruiting methyltransferase SUV39H1. *J Biol Chem* 289: 29691-29700

Shoaib M, Chen Q, Shi X, Nair N, Prasanna C, Yang R, Walter D, Frederiksen KS, Einarsson H, Svensson JP *et al* (2021) Histone H4 lysine 20 mono-methylation directly facilitates chromatin openness and promotes transcription of housekeeping genes. *Nat Commun* 12: 4800

Sun L, Huang Y, Wei Q, Tong X, Cai R, Nalepa G, Ye X (2015) Cyclin E-CDK2 protein phosphorylates plant homeodomain finger protein 8 (PHF8) and regulates its function in the cell cycle. *J Biol Chem* 290: 4075-4085

Tiberi L, Vanderhaeghen P, van den Aemele J (2012) Cortical neurogenesis and morphogens: diversity of cues, sources and functions. *Curr Opin Cell Biol* 24: 269-276

Yao Z, van Velthoven CTJ, Kunst M, Zhang M, McMillen D, Lee C, Jung W, Goldy J, Abdelhak A, Aitken M *et al* (2023) A high-resolution transcriptomic and spatial atlas of cell types in the whole mouse brain. *Nature* 624: 317-332

Zhang I, Boezio GLM, Cornwall-Scoones J, Frith T, Finnie E, Luo J, Jiang M, Howell M, Lovell-Badge R, Sagner A *et al* (2025) The cis-regulatory logic integrating spatial and temporal patterning in the vertebrate neural tube. *Dev Cell*. S1534-5807(25)00407-1

Figure for referee with unpublished data and its description has been removed upon request by the authors.

Dear Dr. Martínez-Balbas,

Thank you for the submission of your revised manuscript to our editorial offices. I have now received the reports from the three referees that I asked to re-evaluate the study, you will find below. As you will see, the referees now fully support publication of your study in EMBO reports. Referee #3 has a request and suggestions to improve the manuscript, I ask you to address in a final revised manuscript. Please also provide a final p-b-p-response to the referee points and the editorial requests below.

Editorial requests:

- Please format of the author list on the title page of the manuscript text in the same way as in the submission system (First Name, Last Name). Moreover, there is a name discrepancy. It is Carballeira in the manuscript vs. Carballeria in the system. Please check and make sure the same correct name is provided.

- Please order the sections like this using these names:

Title - Abstract - Keywords - Introduction - Results - Discussion - Methods - Data availability section (DAS) - Acknowledgements - Disclosure and Competing Interests Statement - References - Figure legends - Expanded View Figure legends

- Please check again that the number "n" for how many independent experiments were performed, their nature (biological versus technical replicates), the bars and error bars (e.g. SEM, SD) and the test used to calculate p-values is indicated in the respective figure legends (main, EV and Appendix figures). Please also check that all the p-values are explained in the legend, and that these fit to those shown in the figure. Please provide statistical testing where applicable. Please avoid the phrase 'independent experiment' but clearly state if these were biological or technical replicates. Please also indicate (e.g. with n.s.) if testing was performed, but the differences are not significant. In case n=2, please show the data as separate datapoints without error bars and statistics. See also:

<http://www.embopress.org/page/journal/14693178/authorguide#statisticalanalysis>

If n<5, please show single datapoints for diagrams. Moreover:

- Please note that the exact p values are not provided in the legends of figures 1b-d; 2f-h; 4a, b, d, e, f; 5a, b, c, d; 6a, b; EV 1a-d; EV 2d, g; EV 3c, e, f

- Please indicate the statistical test used for data analysis in the legends of figures 2a-e; 3c, d, f; EV 2a, f

- Please note that information related to n is missing in the legends of figures 2a, h; EV 2d

- Please note that the error bars are not defined in the legends of figures 5d

- Please note that the measure of center for the error bars needs to be defined in the legends of figures 2h; 4a, d, e, f; 5a, b; EV 1a-d; EV 2d

- Please note that the abbreviations 'CP, IZ, SVZ, VZ' are not defined in the legends of figures 6a, b; EV 3a, d. This needs to be rectified.

- Please note that the white arrows heads are not defined in the legends of figures 6a, b. This needs to be rectified.

- Please add scale bars of similar style and thickness to all microscopic images, using clearly visible black or white bars (depending on the background). Please place these in the lower right corner of the images themselves. Please do not write on or near the bars in the image but define the size in the respective figure legend. Presently, most scale bars have rather illegible text nearby. Please check.

- Please make sure that all the funding information is also entered into the online submission system and that it is complete and similar to the one in the acknowledgement section of the manuscript text file. Presently, these grants are missing in the submission system: 2021AEP079 from the CSIC, Ramon y Cajal fellowship (RYC2022-035576-I), FEDER. Please check.

- Please remove the author list from the Appendix. It is sufficient to state 'Appendix for ...' followed by the title and the Table of contents (TOC). Moreover, please move the legend below each Appendix figure. This is more comprehensible.

- Please add the primer information (Appendix Table S3) to the Reagents & Tools Table and remove the table from the Appendix. Please update the Appendix TOC and any call outs.

- Please add the word Appendix to the callouts of the Appendix figures in the Reagents & Tools Table.

- Please do not add external links regarding the dataset used for Fig. 1A in the legend or the text. Please cite this as formal data citation. See the section 'Data Citation' here:

<https://link.springer.com/journal/44319/submission-guidelines#cms-Reference-guidelines>

- Please also remove now the referee tokens from the data availability section. Please also add direct links to access the GEO dataset series and make sure that all these datasets are public latest at the online publication data of the paper.

- Thanks for providing the source data. Please upload this as one folder per main figure, grouping together all the files for this figure in separate files for each panel (and ZIPed together), and one folder for the EV Figures, grouping together all the files for each Figure in separate folders (and ZIPed together).

- During our image integrity check we noted a reuse of Western blot panels between Fig. 5A and Appendix Fig. S7A and Fig. 5C and Appendix Fig. S7B. Please check. If this reuse is intentional, please clearly indicate this in the respective figure legends.

Please confirm that for all Western blot panels (main, EV, or Appendix figures) the loading control was run on the same gel as the other proteins detected. Please note that we discourage comparisons between samples on different gels/blots, even if the samples derive from one experiment, as confounding factors reduce comparability. If unavoidable, the figure legend must state that the samples derive from the same experiment and that gels/blots were processed in parallel. If a 'representative' loading control is shown for multiple gels/blots, the intra-gel controls should be shown in the source data files, and the figure legends should describe the data displayed accurately. See our author guidelines:

<https://link.springer.com/journal/44319/submission-guidelines#cms-Figure-and-data-presentation> (section 'Electrophoretic gels and blots').

In addition, I would need from you uploaded separately:

I look forward to seeing the further revised version of your manuscript when it is ready. Please let me know if you have questions regarding the revision.

Best,

Referee #1:

My comments have been properly addressed and from my point of view the manuscript is ready for publication.

Referee #2:

The authors have substantially revised the manuscript by performing new experiments and providing necessary information that was previously not included. I support publication of this manuscript in EMBO reports.

Referee #3:

The authors addressed in their revision most of my comments, and I am happy to see the manuscript's improvements. Still, I have some minor concerns, that should be fixed. One of my advises is to go through some phrasings, as you will see by some of the points I am listing - however, I feel that the authors have to go through the written manuscript; there might be other mistakes than the ones I have spotted.

1. There is still no documentation of the bioinformatics pipeline and settings of analysis. Please document these for reproduction of the data in a repository like Github. Without precise documentation of analysis it will be impossible to reproduce findings - which is against FAIR principles of data storage and sharing.

2. In the abstract the following sentence is misleading, as it makes a connection between the different processes through the hyphen "-". Please make sure to refer to the biological processes impacted by Phf8 KD as separate entities; because if a cell suffers DNA damage it does not necessarily correlate directly to a hindered vesicle formation (and the data are not showing this causative connection):

"Loss of PHF8 disrupts amino acid metabolism, blocks autophagy, and hinders vesicle formation-ultimately causing replication defects, DNA damage, and proliferation arrest."

3. Results:

- what does it mean "Phf8 is actively expressed"? How would it look like if a gene is inactively expressed?
- "consist of progenitors, radial glial cells." Maybe insert an "i.e." or "namely" to make it a sentence
- "knock-down PHF8 on NSCs" better: "in NSCs" - and maybe shorten the sentence and split into two sentences.
- "increase in H4K20me1 bulk methylation upon PHF8 depletion (Iacobucci et al, 2021) and (Fig. EV1B)." The last parentheses should follow a full sentence.
- "Interestingly, we observed a decrease on H4K20me1 levels at the SBP gene bodies" better: of H4K20me1 levels
- "that regions with decreased H3K9me3 were significantly enriched at promoters, particularly" - regions mainly classified as promoters
(how can regions enrich at promoters, since promoters are regions?)
- "This assay assesses chromatin accessibility and provide insights", provides
- "NFIA, followed by ATF4, is the most highly expressed transcription factors and is present in the largest number of cells at developmental stages in which PHF8 is actively expressed - transcription factor, and again: what is active expression compared to inactive? This phrasing is used at multiple times in the manuscript
- ATF4 RNA-seq data (Appendix Table S1) - RNA-seq (and maybe shorten this lengthy sentence)
- "Serine, although is not directly incorporated into nitrogenous bases, it supports nucleotide synthesis in two key ways." Needs to be rephrased
- "Thus, cortical neurogenesis proceeds through both direct neurogenesis from RGCs and indirect neurogenesis via intermediate progenitors (IPs)." Redundant summary of the preceding sentences.

ANSWER TO THE EDITOR AND THE REVIEWERS

EMBOR- EMBOR-2025-61935V1 "Epigenetic Regulation of Serine Biosynthesis by PHF8 During Neurogenesis"

We sincerely appreciate both the editor's and the reviewers' comments. We have now provided an answer to the referee's comments following the suggestions made by the editorial.

We include below a detailed, "point-by-point" response to them. Reviewer's suggestions are printed in Times bold, and our replies are printed in Times.

EDITORIAL REQUESTS:

- Please format of the author list on the title page of the manuscript text in the same way as in the submission system (First Name, Last Name). Moreover, there is a name discrepancy. It is Carballeira in the manuscript vs. Carballeria in the system. Please check and make sure the same correct name is provided.

Response:

The format has been corrected and Carballeira's name has been checked.

- Please order the sections like this using these names:
Title - Abstract - Keywords - Introduction - Results - Discussion - Methods - Data availability section (DAS) - Acknowledgements - Disclosure and Competing Interests Statement - References - Figure legends - Expanded View Figure legends

Response:

The sections have been reordered and renamed

- Please check again that the number "n" for how many independent experiments were performed, their nature (biological versus technical replicates), the bars and error bars (e.g. SEM, SD) and the test used to calculate p-values is indicated in the respective figure legends (main, EV and Appendix figures). Please also check that all the p-values are explained in the legend, and that these fit to those shown in the figure. Please provide statistical testing where applicable. Please avoid the phrase 'independent experiment' but clearly state if these were biological or technical replicates. Please also indicate (e.g. with n.s.) if testing was performed, but the differences are not significant. In case n=2, please show the data as separate datapoints without error bars and

statistics.

See

also:

<http://www.embopress.org/page/journal/14693178/authorguide#statisticalanalysis>

Response:

We have carefully checked the statistics and the number of replicates

If $n < 5$, please show single datapoints for diagrams. Moreover:

- Please note that the exact p values are not provided in the legends of figures 1b-d; 2f-h; 4a, b, d, e, f; 5a, b, c, d; 6a, b; EV 1a-d; EV 2d, g; EV 3c, e, f

Response:

The missing information has been added

- Please indicate the statistical test used for data analysis in the legends of figures 2a-e; 3c, d, f; EV 2a, f

Response:

The statistical tests have been added to the figure legends

- Please note that information related to n is missing in the legends of figures 2a, h; EV 2d

Response:

The information has been provided

- Please note that the error bars are not defined in the legends of figures 5d)

Response:

The error bars have been added

- Please note that the measure of center for the error bars needs to be defined in the legends of figures 2h; 4a, d, e, f; 5a, b; EV 1a-d; EV 2d

Response:

The error bars have been defined. They represent mean \pm standard error of the mean (SEM)

- Please note that the abbreviations 'CP, IZ, SVZ, VZ' are not defined in the legends of figures 6a, b; EV 3a, d. This needs to be rectified.

Response:

The abbreviations have been specified.

- Please note that the white arrows heads are not defined in the legends of figures 6a, b. This needs to be rectified.

Response:

The arrows have been defined.

- Please add scale bars of similar style and thickness to all microscopic images, using clearly visible black or white bars (depending on the background). Please place these in the lower right corner of the images themselves. Please do not write on or near the bars in the image but define the size in the respective figure legend. Presently, most scale bars have rather illegible text nearby. Please check.

Response:

The style has been modified

- Please make sure that all the funding information is also entered into the online submission system and that it is complete and similar to the one in the acknowledgement section of the manuscript text file. Presently, these grants are missing in the submission system: 2021AEP079 from the CSIC, Ramon y Cajal fellowship (RYC2022-035576-I), FEDER. Please check.

Response:

The new funding has been added to the system

- Please remove the author list from the Appendix. It is sufficient to state 'Appendix for ...' followed by the title and the Table of contents (TOC). Moreover, please move the legend below each Appendix figure. This is more comprehensible.

Response:

The author list has been eliminated

- Please add the primer information (Appendix Table S3) to the Reagents & Tools Table and remove the table form the Appendix. Please update the Appendix TOC and any call outs.

Response:

Appendix Table S3 has been removed, and the primer information has been moved to the Reagents section

- Please add the word Appendix to the callouts of the Appendix figures in the Reagents & Tools Table.

Response:

The word Appendix is used to callouts The Appendix figures

- Please do not add external links regarding the dataset used for Fig. 1A in the legend or the text. Please cite this as formal data citation. See the section 'Data Citation' here: <https://link.springer.com/journal/44319/submission-guidelines#cms-Reference-guidelines>

Response:

The link has been eliminated

- Please also remove now the referee tokens from the data availability section. Please also add direct links to access the GEO dataset series and make sure that all these datasets are public latest at the online publication data of the paper.

Response:

The number of referee tokens has been removed and the links to access the GEO dataset series have been added

- Thanks for providing the source data. Please upload this as one folder per main figure, grouping together all the files for this figure in separate files for each panel (and ZIPed together), and one folder for the EV Figures, grouping together all the files for each Figure in separate folders (and ZIPed together).

Response:

The data source has been submitted in accordance with the instructions.

- During our image integrity check we noted a reuse of Western blot panels between Fig. 5A and Appendix Fig. S7A and Fig. 5C and Appendix Fig. S7B. Please check. If this reuse is intentional, please clearly indicate this in the respective figure legends.

Response:

In all cases, the samples correspond to those derived from the same experiment and were probed with different antibodies. To avoid confusion and because it was unnecessary, the panel in Fig. S7A has been removed. The reason for reusing the control in Fig. S7B has been indicated in the figure legend

- Please confirm that for all Western blot panels (main, EV, or Appendix figures) the loading control was run on the same gel as the other proteins detected. Please note that we discourage comparisons between samples on different gels/blots, even if the samples derive from one experiment, as confounding factors reduce comparability. If unavoidable, the figure legend must state that the samples derive from the same experiment and that gels/blots were processed in parallel. If a 'representative' loading control is shown for multiple gels/blots, the intra-gel controls should be shown in the source data files, and the figure legends should describe the data displayed accurately. See our author guidelines:

<https://link.springer.com/journal/44319/submission-guidelines#cms-Figure-and-data-presentation> (section 'Electrophoretic gels and blots').

Response:

Comparisons were made between bands identified on the same gel

In addition, I would need from you uploaded separately:

- a short, two-sentence summary of the manuscript (not more than 35 words).**
- two to four short bullet points highlighting the key findings of your study (two lines each).**
- a schematic summary figure as separate file that provides a sketch of the major findings**

Response:

The summary, bullet points, and synopsis figure have been uploaded separately.

RERESPONSE TO REVIEWERS

Referee #1:

My comments have been properly addressed and from my point of view the manuscript is ready for publication.

Response:

We express our gratitude to Reviewer 1 for providing constructive and positive feedback, which I am confident has helped improve the quality of our work.

Referee #2:

The authors have substantially revised the manuscript by performing new experiments and providing necessary information that was previously not included. I support publication of this manuscript in EMBO reports.

Response:

I would like to express my gratitude to Referee 2 for their constructive comments, which have prompted us to reflect on certain aspects of our work. I am confident that these comments have contributed to the overall improvement of our work, and we wish to extend our thanks.

Referee #3:

The authors addressed in their revision most of my comments, and I am happy to see the manuscript's improvements. Still, I have some minor concerns, that should be fixed. One of my advises is to go through some phrasings, as you will see by some of the points I am listing - however, I feel that the authors have to go through the written manuscript; there might be other mistakes than the ones I have spotted.

Response:

I would like to thank Referee 3 for their constructive feedback. I am confident that their comments have helped improve the overall quality of the manuscript, and we sincerely appreciate their valuable contribution.

1. There is still no documentation of the bioinformatics pipeline and settings of analysis. Please document these for reproduction of the data in a repository like Github. Without precise documentation of analysis it will be impossible to reproduce findings - which is against FAIR principles of data storage and sharing.

Response:

Following Referee 3's request, we have created a Python script implementing the ChIP-seq pipeline and have uploaded it to GitHub. https://github.com/ClinicalTranslationalBioinformatics/artes_et_al_2025. The bioinformatic analysis of the RNA-seq data performed by BGI has now been clearly explained and clarified in the revised manuscript.

2. In the abstract the following sentence is misleading, as it makes a connection between the different processes through the hyphen "-". Please make sure to refer to the biological processes

impacted by Phf8 KD as separate entities; because if a cell suffers DNA damage it does not necessarily correlate directly to a hindered vesicle formation (and the data are not showing this causative connection): "Loss of PHF8 disrupts amino acid metabolism, blocks autophagy, and hinders vesicle formation-ultimately causing replication defects, DNA damage, and proliferation arrest."

Response:

The referee is absolutely correct. Following their suggestion, we have modified the sentence in the abstract.

3. Results:

• **what does it mean "Phf8 is actively expressed"? How would it look like if a gene is inactively expressed?**

Response:

The expression has been modified, and the word "actively" has been removed.

• **"consist of progenitors, radial glial cells." Maybe insert an "i.e." or "namely" to make it a sentence**

Response:

We have added the word "namely."

• **"knock-down PHF8 on NSCs" better: "in NSCs" - and maybe shorten the sentence and split into two sentences.**

Response:

The sentence has been revised.

• **"increase in H4K20me1 bulk methylation upon PHF8 depletion (Iacobucci et al, 2021) and (Fig. EV1B)." The last parentheses should follow a full sentence.**

Response:

The error has been rectified.

• **"Interestingly, we observed a decrease on H4K20me1 levels at the SBP gene bodies" better: of H4K20me1 levels**

Response:

“On” has been replaced with “of”

- **"that regions with decreased H3K9me3 were significantly enriched at promoters, particularly"**
- regions mainly classified as promoters (how can regions enrich at promoters, since promoters are regions?)

Response:

The sentence has been revised

- **"This assay assesses chromatin accessibility and provide insights", provides**

Response:

The mistake has been corrected.

- **"NFIA, followed by ATF4, is the most highly expressed transcription factors and is present in the largest number of cells at developmental stages in which PHF8 is actively expressed - transcription factor, and again: what is active expression compared to inactive? This phrasing is used at multiple times in the manuscript**

Response:

The term “actively” has been removed throughout the manuscript.

- **ATF4 RNA-seq data (Appendix Table S1) - RNA-seq (and maybe shorten this lengthy sentence)**

Response:

The sentence has been modified

- **"Serine, although is not directly incorporated into nitrogenous bases, it supports nucleotide synthesis in two key ways." Needs to be rephrased**

Response:

The sentence has been revised

- **"Thus, cortical neurogenesis proceeds through both direct neurogenesis from RGCs and indirect neurogenesis via intermediate progenitors (IPs)." Redundant summary of the preceding sentences.**

Response:

The sentence has been removed

Dr. Marian Martínez-Balbas
CSIC
Structural and Molecular Biology
Baldiri i Reixac 15-21
Barcelona, Barcelona 08028
Spain

Dear Dr. Martínez-Balbas,

Thank you for the submission of your final revised manuscript to our editorial offices. It now went through this and your final p-b-p-response and consider the remaining points of referee #3 and the editorial requests as adequately addressed.

I am thus very pleased to accept your manuscript for publication in the next available issue of EMBO reports. Thank you for your contribution to our journal.

You may qualify for financial assistance for your publication charges - either via a Springer Nature fully open access agreement or an EMBO initiative. Check your eligibility: <https://link.springer.com/journal/44319/how-to-publish-with-us>

Yours sincerely,

>>> Please note that it is EMBO Reports policy for the transcript of the editorial process (containing referee reports and your response letter) to be published as an online supplement to each paper. If you do NOT want this, you will need to inform the Editorial Office via email immediately. More information is available here: <https://link.springer.com/partners/embo-press/editorial-policies#Peer%20review>